# BACKPROPAGATION-FREE TRAINING OF NEURAL PDE SOLVERS FOR TIME-DEPENDENT PROBLEMS

## ABSTRACT

Approximating solutions to time-dependent Partial Differential Equations (PDEs) is one of the most important problems in computational science. Neural PDE solvers have shown promise recently because they are mesh-free and easy to implement. However, backpropagation-based training often leads to poor approximation accuracy and long training time. In particular, capturing high-frequency temporal dynamics and solving over long time spans pose significant challenges. To address these, we present an approach to training neural PDE solvers without backpropagation by integrating two key ideas: separation of space and time variables and random sampling of weights and biases of the hidden layers. We reformulate the PDE as an Ordinary Differential Equation (ODE) using a neural network ansatz, construct neural basis functions only in the spatial domain, and solve the ODE leveraging classical ODE solvers from scientific computing. We demonstrate that our backpropagation-free algorithm outperforms the iterative, gradient-based optimization of physics-informed neural networks with respect to training time and accuracy, often by 1 to 5 orders of magnitude using different complicated PDEs characterized by high-frequency temporal dynamics, long time span, complex spatial domain, non-linearities, shocks, and high dimensionality.

## 1 INTRODUCTION

Approximating solutions of partial differential equations (PDEs) is vital in computational science and engineering. Traditional mesh-based numerical methods like finite differences, finite volumes, finite elements, or mesh-free methods based on global basis functions like spectral methods have been developed for decades. These methods often approximate PDE solutions with high accuracy and are grounded in theory. However, mesh-based methods are often difficult to implement on complicated domains due to the meshing difficulties and can be prohibitively expensive for high-dimensional problems owing to the curse of dimensionality. Traditional spectral methods often struggle with complicated domains and locally sharp gradients in the PDE solution (Boyd, 2001).

Deep neural networks have recently shown significant promise for approximating solutions of PDEs because of the mesh-free construction of basis functions, high expressivity of neural networks (Rudi & Rosasco, 2021), their ability to represent functions in high dimensions (E, 2020; Wu & Long, 2022), powerful software for automatic differentiation (e.g., Pytorch (Paszke et al., 2017), TensorFlow (Abadi et al., 2015), and specialized software like DeepXDE (Lu et al., 2021b)). Earlier work on solving PDEs using neural networks (Dissanayake & Phan-Thien, 1994; Lagaris et al., 1998) was recently popularized in the form of Physics-informed neural networks (PINNs) and neural operators by Raissi et al. (2019); Lu et al. (2021a); Li et al. (2020); Raonic et al. (2024); Han et al. (2018), and Sirignano & Spiliopoulos (2018). However, numerous challenges are becoming apparent. Next, we outline some of the key drawbacks of existing neural PDE solvers based on gradient-based iterative optimization that motivate our work: training difficulties, capturing high-frequency temporal dynamics, as well as long training time and low accuracy.

**Training difficulties:** Neural PDE solvers that rely on backpropagation-through-time require computing gradients of the loss function with respect to the network parameters along trajectories (Um et al., 2020). This usually leads to challenges posed by exploding and vanishing gradients during the iterative, gradient-based training procedure Pascanu et al. (2013); Schmidt et al. (2019). Physics-informed neural networks and their variants are not iterative in the same way, but require minimizing

a loss function involving the PDE residual, boundary conditions, and initial conditions. Rathore et al. (2024) demonstrate that the PDE residual loss primarily causes ill-conditioning of the PINN loss. It has been shown that even in simple settings, the PINN loss is very challenging to minimize using backpropagation (Krishnapriyan et al., 2021; Wang et al., 2021; 2022). Though various approaches such as balancing different loss terms (Yao et al., 2023), regularization (Lu et al., 2021c; Yu et al., 2022), and different optimizers (Müller & Zeinhofer, 2023; Liu et al., 2024) were introduced to alleviate some of the problems, it is still quite difficult to optimize PINNs with backpropagation.

**Capturing high-frequency temporal dynamics and solving PDEs over a long time span:** The temporal structure of initial value PDEs is local, as each subsequent step depends solely on the values of the preceding spatial slice — a property that typical physics-informed-machine-learning-based approaches, which treat time similar to an extra spatial dimension, fail to consider. We show that by using neural basis functions only in space and classical ODE solvers in time, one can capture high-frequency temporal dynamics and solve PDEs over long time spans.

**Long training time and low accuracy:** Approximation errors tend to be significant because of the gradient-based iterative optimization of network parameters and the challenges associated with handling time as an additional spatial dimension. This often leads to longer training times and much lower accuracy than classical mesh-based methods, especially for problems involving complex temporal dynamics or long time spans.

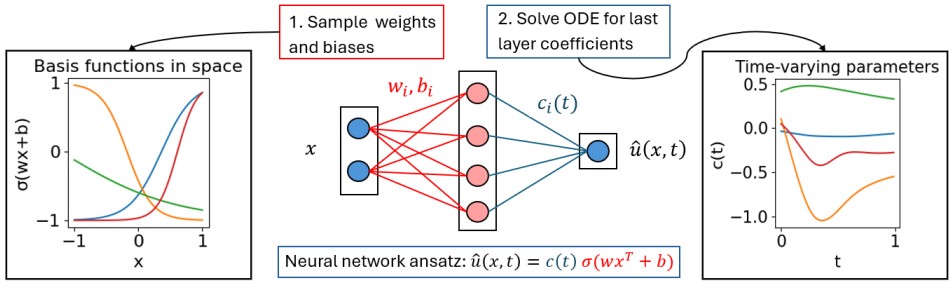

Figure 1: Solving a time-dependent PDE with a neural network ansatz: We sample hidden layer parameters $w$, $b$ and fix the neural spatial basis functions (left) $\phi_i = \sigma(w_i x + b_i)$. The PDE to be solved is reformulated as an ODE in terms of the time-dependent output layer coefficients $c_i(t)$ (right), obtained by solving the ODE and computing the solution $\hat{u}(x,t)$.

To address these limitations, we propose an approach to training neural PDE solvers without backpropagation by integrating two key ideas: separation of variables (space and time) and random sampling of hidden layer weights and biases. For the latter, we employ Extreme Learning Machines (ELMs, cf. Huang et al. (2006)) and Sampling Where It Matters (SWIM, cf. Bolager et al. (2023)). Figure 1 illustrates the key components of our approach. Our key contributions are listed below.

- We propose two approaches to solving time-dependent PDEs with neural networks without backpropagation that synergistically combine data-driven (SWIM-ODE) or data-agnostic (ELM-ODE) sampling algorithms for computing hidden layer parameters with classical ODE solvers from scientific computing (See Section 3.2, Section 3.3).

- We propose novel techniques to satisfy boundary conditions for solving time-dependent PDEs with a neural network ansatz (See Section 3.4).

- We demonstrate the strengths of our backpropagation-free training algorithm for neural PDE solvers—high accuracy, reduced training time, spectral convergence, and mesh-free basis functions—by solving complex PDEs involving high-frequency temporal dynamics, long-time simulations, complicated domains, non-linearities, shocks, and high-dimensionality (See Section 4).

Our approach outperforms PINNs trained with backpropagation by 1-5 orders of magnitude in accuracy and up to 4 orders of magnitude in training time. Compared to classical mesh-based methods like FEM, our approach is very easy to implement on complicated geometries. It yields a comparable performance (in low dimensions), but it can also deal with high-dimensional PDEs, where mesh-based methods suffer from the curse of dimensionality.

## 2 RELATED WORK

**Randomized neural networks for solving PDEs** have been studied, mostly combining Extreme Learning Machines (ELMs) with the self-supervised setting of PINNs (Chen et al., 2024; Wang & Dong, 2024; Shang & Wang, 2024; Sun et al., 2024). Dwivedi & Srinivasan (2020) propose a physics-informed extreme learning machine (PIELM) to efficiently solve linear PDEs, while Calabrò et al. (2021); Galaris et al. (2022) employ ELMs to learn invariant manifolds as well as PDE from data. Dong & Yang (2022) establish that given a fixed computational budget, ELMs achieve substantially higher accuracy compared to classical second-order FEM and slightly higher accuracy compared to higher-order FEM. For static, nonlinear PDEs, ELMs can be used together with nonlinear optimization schemes (Fabiani et al., 2021). On larger spatiotemporal domains, Dong & Li (2021) and Dwivedi et al. (2021) propose using multiple distributed ELMs on multiple subdomains. These approaches treat time similarly to an extra dimension in space, and neural basis functions are used to span (a part of) the entire spatiotemporal domain, unlike our approach.

**Neural Galerkin schemes** (cf. work from Finzi et al. (2023); Aghili et al. (2024); Berman et al. (2024); Bruna et al. (2024)) offer an alternative to the full spatiotemporal approach of the randomized neural networks and PINNs. These approaches treat all or sparse subsets of network parameters, beyond just the last layer's parameters, as time-dependent. This leads to a much larger system of ODEs compared to our approach. Chen et al. (2023); Yin et al. (2023) also use neural network basis functions to represent the space component but are based on backpropagation.

**Physics-informed neural networks (PINNs)** are widely used to solve PDEs with neural networks. For high-frequency temporal variations, Krishnapriyan et al. (2021) propose curriculum learning with gradually increasing advection coefficients. Our approach is much easier to implement, much more computationally efficient, and accurate, as we demonstrate in Section 4.1. Subramanian et al. (2023) propose using adaptive self-supervision of PINNs for sampling collocation points using the gradient of the loss function. We instead use the solution gradient to capture locally sharp features in the solution (cf. Section 4.4). Many specialized approaches based on PINNs (cf Cho et al. (2024), Meng et al. (2020)), Sharma & Shankar (2022), and Chiu et al. (2022)), methods based on hash-encoding (cf. Huang & Alkhalifah (2024), Wang et al. (2024a)) and transfer learning (cf. Kapoor et al. (2024b))) have been proposed, but are still based on backpropagation, unlike ours.

**Classical numerical methods to solve PDEs:** Finite elements, finite volumes, and finite differences have been used to solve PDEs for decades. They often have a rich theoretical grounding and high accuracy. Isogeometric analysis (IGA) is such a method in which spline-based basis functions are defined over a structured grid (cf. Hughes et al. (2005); Cottrell et al. (2009; 2006)). Mesh-based methods often entail a time-consuming setup phase, especially when mesh generation is challenging. In this work, we benchmark our results against IGA and finite-element-based methods.

For an extended review of related work, please refer to Appendix A.

## 3 APPROXIMATION OF PDE SOLUTIONS USING NEURAL NETWORKS

We discuss solution methods for PDEs on domains $\Omega \in \mathbb{R}^d$ with boundary $\partial\Omega$. We address linear and nonlinear time-dependent PDEs with solutions $u : \Omega \times \mathbb{R} \to \mathbb{R}$. These PDEs are defined by linear operators $\mathcal{L}$ and $\mathcal{B}$ that only involve derivative operators in space, and functions $f : \Omega \to \mathbb{R}$, $g : \partial\Omega \to \mathbb{R}$, and $u_0 : \Omega \to \mathbb{R}$ that define forcing, boundary condition, and initial condition, respectively. For nonlinear PDEs, we denote the nonlinear operator by $\mathcal{N}$, and its scaling $\gamma \geq 0$ is either zero (for linear PDEs) or positive (for nonlinear PDEs). Then,

$$u_t(x,t) + \mathcal{L}u(x,t) + \gamma\mathcal{N}(u)(x,t) = f(x), \ x \in \Omega, \ t \in [0,T], \tag{1}$$

$$\mathcal{B}u(x,t) = g(x), \ x \in \partial\Omega, \quad u(x,0) = u_0(x), \ x \in \Omega, \tag{2}$$

where we denote by $u_t$ the first derivative of $u$ by time.

We first describe the neural network ansatz in Section 3.1, and then describe how to construct the spatial basis functions of the ansatz in Section 3.2. For time-dependent PDEs, we propose solving an ordinary differential equation associated with our construction of the spatial basis using classical ODE solvers in Section 3.3 by evolving the last layer coefficients in time. In Section 3.4, we explain different approaches for satisfying boundary conditions by adding a linear layer. Lastly, in Section 3.5, we summarize the algorithm for backpropagation-free training of neural PDE solvers.

## 3.1 Neural network ansatz

We parameterize the approximation of a solution with a neural network with one hidden layer, activation function $\sigma = \tanh$, $M$ neurons, so that

$$\hat{u}(x,t) = C(t)[\Phi(x), \mathbb{1}] = c(t)\sigma(Wx^\top + b) + c_0(t). \tag{3}$$

Here, $c(t) \in \mathbb{R}^{1 \times M}$ and $c_0(t) \in \mathbb{R}$ are time-dependent parameters, $W \in \mathbb{R}^{M \times d}$ and $b \in \mathbb{R}^{M \times 1}$ are time-independent parameters, and $C := [c, c_0] \in \mathbb{R}^{1 \times (M+1)}$. The activation functions are stacked in $\Phi = [\phi_1, \ldots, \phi_M]$, where $\phi_m(x) = \sigma(w_m x^\top + b_m)$. We will distinguish between two approaches with different weight spaces for the hidden layer. For the extreme learning machine (ELM) framework, the weight and bias space is the full space $\mathbb{R}^{M \times d} \times [-\eta, \eta]$, where $\eta$ is sufficiently large. The second approach, the sampling where it matters (SWIM) framework, follows Bolager et al. (2023) and restricts the weight space to $\Omega \times \Omega$. We construct each weight and bias pair $w_m, b_m$ by taking two points $x^{(1)}, x^{(2)} \in \Omega$ and construct the weight and bias as $w_m = s_1 \frac{x^{(2)} - x^{(1)}}{\|x^{(2)} - x^{(1)}\|^2}$, $b_m = -\langle w_m, x^{(1)} \rangle + s_2$, where $s_1, s_2$ are constants dependent on the activation function. We distinguish between the two approaches by referring to neural networks constructed by SWIM or ELM.

## 3.2 Computing hidden layer parameters without gradient-based optimization

To sample the coefficients of the first hidden layer, we propose two approaches: ELM and SWIM.

**ELM (Data-agnostic)**: In ELM, the weights are sampled from a Gaussian distribution, and biases are sampled from a uniform distribution in $[-\eta, \eta]$ for each hidden layer, where $\eta$ is a hyper-parameter.

**SWIM (Data-dependent)**: The SWIM algorithm samples weights and biases using a data-dependent distribution. The weight and bias of each neuron in the hidden layer are sampled using one pair of spatial collocation points $(x^{(1)}, x^{(2)})$. In the unsupervised setting, one can choose pairs of collocation points from a uniform distribution over all possible pairs of collocation points, which is the default setting in this paper, as we do not know the solution of the PDE beforehand. In the supervised setting, the data points are selected based on the density $\frac{\|f(x^{(2)}) - f(x^{(1)})\|}{\|x^{(2)} - x^{(1)}\|}$, with $f$ being the true function in a supervised setting. The weights and bias of each neuron with a $\tanh$ activation function are chosen such that the neuron's output is $-0.5$ for the input $x^{(1)}$ and $+0.5$ for the input $x^{(2)}$. This ensures that the centers of the activation functions are always placed in the spatial domain—unlike ELM, where the centers of the functions could be randomly placed outside the spatial domain. It also ensures that the activation functions are oriented in the direction from point $x^{(1)}$ to point $x^{(2)}$.

The key benefits of randomly sampling basis functions include much shorter training times and improved accuracy compared to PINNs (both from one to five orders of magnitude), nearly matching the numerical state-of-the-art solvers. Moreover, the advantages compared to the classical numerical solvers such as finite elements, finite differences, or finite volume approaches include spectral convergence (i.e., requiring much fewer basis functions) without requiring a mesh, making it much easier to implement on complex geometries.

The suitability of each of the proposed approaches depends on the true PDE solution's gradient distribution. For a detailed comparison, please refer to Appendix B.1.3. We empirically observe that ELM performs better in approximating solutions with shallow gradients, while SWIM (by sampling weights from close data points) performs better in approximating solutions with steep gradients. In Figure 2, we illustrate the difference between the basis functions sampled with ELM and SWIM.

## 3.3 Solving time-dependent PDEs by separation of variables

For both linear and nonlinear time-dependent PDEs, we plug the ansatz (see Equation (3)) into the PDE Equation (1) to re-formulate it as an ODE for the time-dependent coefficients $c(t)$. We first assemble $N_c$ spatial collocation points in $\mathbb{R}^{1 \times d}$ in the columns of a matrix $X \in \mathbb{R}^{N_c \times d}$. We next sample weights and biases of $M$ neurons and evaluate $\Phi(X)$. We then reformulate the PDE described in Equation (1) as an ODE,

$$C_t(t) = R(X, C(t))[\Phi(X), \mathbb{1}]^+, \quad \text{where}$$

$$R(X, C(t)) = -C(t)\mathcal{L}[\Phi(X), \mathbb{1}] - \gamma \mathcal{N}(C(t)[\Phi(X), \mathbb{1}]) + [f(X)]^\top, \tag{4}$$

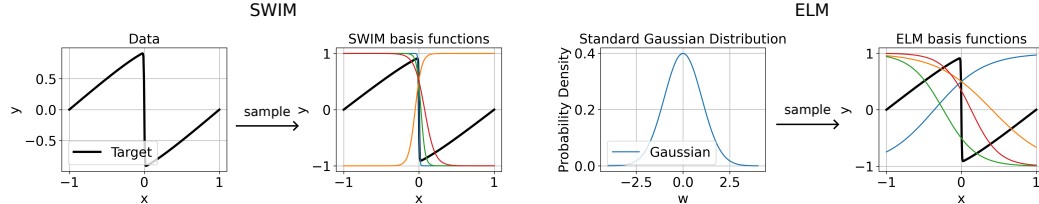

Figure 2: SWIM sampling (left) is data-dependent and allows placement of basis functions near steep gradients. ELM sampling (right) is data-agnostic because the parameters of the basis functions are sampled from a Gaussian distribution.

where we denote the pseudo-inverse as $\cdot^+$, $[\Phi(X), \mathbb{1}] \in \mathbb{R}^{(M+1) \times N_c}$. The initial condition for this ODE is given through $C(0) = u(X, 0)^\top [\Phi(X), \mathbb{1}]^+$. We use classical solvers with step-size control like the Runge-Kutta-45 method (cf. Dormand & Prince (1980)) and implicit ODE solvers like LSODA (cf. Petzold (1983)). We then interpolate the predicted solution $C(t)$ at test points. The ansatz (see Equation (3)) does not explicitly take boundary conditions into consideration. In the next section, we discuss how to address this.

### 3.4 APPROACHES FOR SATISFYING BOUNDARY CONDITIONS

To satisfy certain boundary conditions, we propose adding a linear transformation $A \in \mathbb{R}^{M_b \times M_s}$, where $M_s := M$. We call this a "boundary-compliant layer" (See Figure 3). With this linear transformation, we now rewrite Equation (4) to

$$C_t(t) = R(X, C(t))\Phi_A(X)^+, \quad \text{where}$$
$$R(X, C(t)) = -C(t)\mathcal{L}\Phi_A(X) - \gamma \mathcal{N}(C(t)\Phi_A(X)) + [f(X)]^\top, \tag{5}$$

and $\Phi_A := [A\Phi, \mathbb{1}]$ and $C(t) \in \mathbb{R}^{1 \times (M_b+1)}$. The boundary conditions are dictated by $\mathcal{B}$ and $g$, which alters how we construct $A$. We now discuss how to compute $A$ for different boundary conditions.

**Periodic boundary condition:** If each basis function satisfies the periodic boundary condition, then the ansatz, a linear combination of these functions, will also satisfy it. Thus, we find $A$ so that $A\Phi(x_l) = A\Phi(x_r)$, where $x_l, x_r$ are the left and right boundary points of the spatial domain. In this paper, if required for a given PDE, for $x \in \Omega$ and $k = 1, 2, \ldots, M_s$, we approximate $[A\Phi]_k(x) = \sin(kx)$ (for $k$ even) and $[A\Phi]_k(x) = \cos(kx)$ (for $k$ odd) and set $c_0(t) = 1$ for all $t$. This can be useful for PDEs where the basis functions are not known explicitly but only through boundary conditions, which we can then incorporate by constructing useful outer basis functions.

**Dirichlet boundary conditions:** For zero Dirichlet boundary condition $u(x) = 0$, we can use the technique described above by choosing basis functions so that $A\phi(x) = 0$ for $x \in \partial\Omega$.

For non-zero Dirichlet boundary condition, where $u(x) = g(x)$, we augment the ODE (Equation (5)) with an additional equation, $\hat{u}_t(x) = -\kappa(\hat{u}(x) - g(x))$ for $x \in \partial\Omega$, and solve the augmented ODE

$$C_t(t) = \underbrace{[R(X, C(t)), -\kappa(C(t)\Phi_A(X_b) - g(X_b)^\top)]}_{\in \mathbb{R}^{1 \times (N_c + N_b)}} \underbrace{\Phi_A([X, X_b])^+}_{\in \mathbb{R}^{(N_c + N_b) \times (M_b+1)}},$$

where $\kappa > 0$ is a fixed parameter, $X$ are the $N_c$ collocation points and $X_b \in \mathbb{R}^{N_b \times d}$ is a collection of $N_b$ points on the boundary $\partial\Omega$. The intuition behind the augmented ODE for the boundary points is that the approximate solution is forced towards the true solution on the boundary with a rate proportional to the difference $(\hat{u}(x, t) - g(x))$ at any time step. We choose a default value of $\kappa = 100$. This technique with the augmented ODE allows setting $A$ to the identity matrix (not using the linear layer at all) to enforce the Dirichlet boundary conditions.

**Other types of boundary conditions:** We use similar ideas to deal with time-dependent Dirichlet and Neumann boundary conditions (See Appendix B.1.1).

### 3.5 SVD Layer and summary of the backpropagation-free algorithm

**SVD layer:** As the last step in the construction of our architecture, we add a linear layer to improve the condition number of the associated ODE in Equation (5) and to reduce the size of the ODE system. To achieve this, we propose orthogonalizing the basis functions using an "SVD layer". We compute a truncated singular value decomposition of $A\Phi(X) \in \mathbb{R}^{M_b \times N_c}$ to obtain matrices $V_r, \Sigma_r$, and $U_r$ with $r \leq M_b$ such that $V_r\Sigma_r U_r^\top = A\Phi(X) + O(\Sigma_{r+1})$. We then define $A_r := V_r^\top A$ and use it instead of the matrix $A$ and $C(t) \in \mathbb{R}^{1 \times (r+1)}$. This ensures $A_r\Phi(X)$ are orthogonal functions on the data $X$, and the matrix $A_r\Phi(X)$ has a bounded condition number. Our ablation study reveals that the SVD layer improves speed (1.2–77x) and reduces the dimension of the ODE system (1.2–22x).

This completes the full procedure and we summarize it in Algorithm 1, where the hyper-parameter $\epsilon_{SVD}$ is a ratio of the largest to smallest singular value that governs how many singular values should be retained (width of the SVD-layer). In addition, Figure 3 visualizes the complete model $\hat{u}$. For details on reformulating PDEs as ODEs, please refer to Appendix B.1.2.

---

**Algorithm 1** Backpropagation-free training algorithm to solve a given PDE

---

**Input:** PDE (Equation (1)) with boundary and initial conditions (Equation (2)), test grid points $X_{\text{test}} \times T_{\text{test}}$
**Output:** Solution of the PDE evaluated on the test grid points $\hat{u}(X_{\text{test}}, T_{\text{test}})$
**Parameters:** $N_c$, $M_s$, $M_b$, $\epsilon_{SVD}$

1: Sample $N_c$ collocation points in a $d$-dimensional space and store it as a matrix $X \in \mathbb{R}^{N_c \times d}$
2: Construct hidden layer parameters $\{w_m, b_m\}_{m=1}^{M_s}$ using SWIM or ELM      ▷ Section 3.2
3: Compute the output of the hidden layer $\Phi(X) \in \mathbb{R}^{M_s \times N_c}$
4: Construct parameters of the linear hidden layer and evaluate $A\Phi(X) \in \mathbb{R}^{M_b \times N_c}$ ▷ Section 3.4
5: Compute truncated SVD with $\epsilon_{SVD}$: $V_r\Sigma_r U_r^\top = A\Phi(X)$ and compute $V_r^\top A\Phi(X) = A_r\Phi(X)$
6: Compute the spatial basis functions $\Phi_{A_r}(X) := (A_r\Phi(X), 1)^\top \in \mathbb{R}^{(r+1) \times N_c}$
7: Compute the initial condition for the last layer parameters: $C(0) = u(X, 0)^\top \Phi_{A_r}(X)^+$
8: Compute $C(t) \in \mathbb{R}^{1 \times (r+1)}$ by solving the ODE using basis functions $\Phi_{A_r}$      ▷ Equation (5)
9: Compute $\hat{u}(X_{\text{test}}, T_{\text{test}}) = C(T_{\text{test}})\Phi_{A_r}(X_{\text{test}})$      ▷ Equation (3)

---

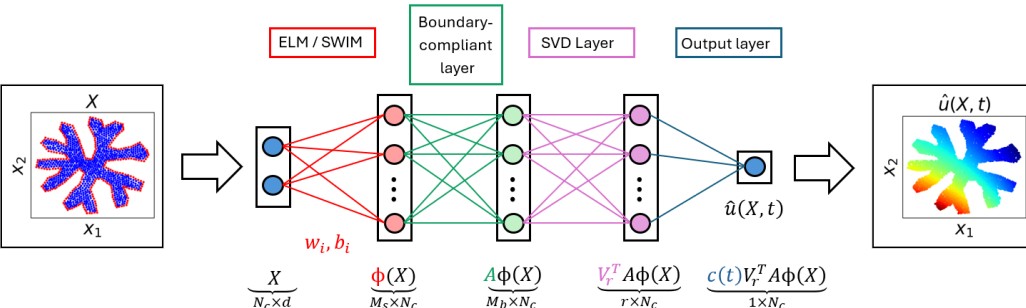

Figure 3: Architecture of our neural-PDE solver trained with backpropagation-free training algorithm.

## 4 Computational experiments

We now demonstrate how our approach of separation of variables can be used to solve several time-dependent PDEs, each involving a different challenge. We compare our approach with physics-informed neural networks (PINNs), both classical Raissi et al. (2019) and causality-respecting (causal PINNs) Wang et al. (2024b). In contrast to the trend of comparing neural PDE-solvers only with other neural PDE-solvers, leading to overly optimistic views of neural solvers and neglecting numerical methods (cf. McGreivy & Hakim (2024)), we also benchmark our method against the state-of-the-art, mesh-based IGA-FEM method Hughes et al. (2005); Cottrell et al. (2009; 2006) or classical FEM. We use the root mean squared error (RMSE) and the relative $L^2$ error to quantify errors in all experiments (cf. Appendix B for the definitions). We compute the test error on a uniform

grid for all time-dependent PDEs with 256 points in space and 100 points in time. We perform all experiments with three seeds and report the mean and standard deviation in all numerical examples. We use the method `solve_ivp` in the Python package SciPy (cf. Virtanen et al. (2020)) to solve ODEs in Equation (5). The software and hardware environments used to perform the experiments are listed in Appendix B. Table 1 lists all PDEs we solve, together with forcing and boundary terms. Please refer to Appendix C for details of the PDEs, a detailed comparison with other approaches, and ablation studies for all computational experiments described below.

Table 1: Summary of PDEs we solve in this paper. $T$ denotes the final time, functions $f, g$ are forcing and boundary terms, and the parameters $\beta, \nu$ are described in their subsections.

| Sec. | PDE | | Boundary | Domain |
|---|---|---|---|---|
| 4.1 | Advection | $u_t + \beta u_x = 0$ | $u(0, t) = u(2\pi, t)$ | $[0, 2\pi] \times [0, T]$ |
| 4.2 | Euler-Bernoulli | $u_{tt} + u_{xxxx} = f(x, t)$ | $(u, u_{xx}) = 0$ | $[0, \pi] \times [0, 1]$ |
| 4.3 | Nonlinear diffusion | $u_t - u\Delta u = f(x, t)$ | $g(x, t)$ | $[0.65, 0.9]^2 \times [0, 1]$ |
| 4.4 | Burgers' | $u_t + uu_x - \nu\Delta u = 0$ | 0 | $[-1, 1] \times [0, 1]$ |
| 4.5 | (n-dim) Diffusion | $u_t - \Delta u = f(x, t)$ | $g(x, t)$ | $[-1, 1]^n \times [0, 1]$ |

### 4.1 HIGH ADVECTION SPEEDS AND LONG-TIME SIMULATION

We consider the linear advection equation $u_t + \beta u_x = 0$ (also see Appendix C.1) with the initial condition $u(x, 0) = \sin(x)$ and periodic boundary conditions. The analytical solution is given by $u(x, t) = \sin(x - \beta t)$.

**High advection speeds:** We solve the advection equation using different neural PDE solvers and IGA-FEM for increasing flow velocities $\beta$ over the domain $\Omega \times T = [0, 2\pi] \times [0, 1]$. The details on hyper-parameters and the setup of this experiment are listed in Appendix C.1. Figure 4 shows that approaches using basis functions in the entire spatiotemporal domain, such as PINNs, ELM, and SWIM, fail as the flow velocity $\beta$ increases beyond 40. In contrast, ELM-ODE, SWIM-ODE, and IGA-FEM can accurately solve the PDE, even for high values of $\beta$. Figure 5 shows that for $\beta = 40$, $L^2_{\text{relative}}$ decays exponentially with the number of basis functions for ELM-ODE, SWIM-ODE, and IGA-FEM. In contrast to IGA-FEM, which uses local basis functions, ELM-ODE and SWIM-ODE require fewer basis functions for a fixed $L^2_{\text{relative}}$ because they use global basis functions. In this example, PINNs yield high errors. Note that compared to the curriculum learning approach proposed by Krishnapriyan et al. (2021), ELM-ODE and SWIM-ODE produce errors that are 4-5 orders of magnitude lower for the advection coefficient $\beta = 40$, are extremely fast. Our approach even works for convection coefficients as high as $10^4$, where traditional neural PDE solvers completely fail.

**Long-time simulation:** We attempt to solve the PDE with $\beta = 1$ for $T = [0, 1000]$, with the true solution shown in Figure 6a. Simulating long-time dynamics is a longstanding challenge for traditional neural PDE solvers (Lippe et al., 2024; Kapoor et al., 2024a), and all solvers using neural network basis functions extending over both space and time, such as PINN, fail at approximating functions on long time intervals. Figure 6b shows that with ELM-ODE and SWIM-ODE, we can solve over 1000 seconds with $L^2_{\text{relative}}$ of less than $0.001\%$, requiring only 0.94 seconds of runtime.

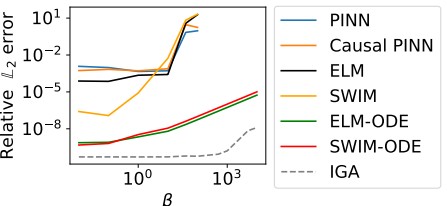

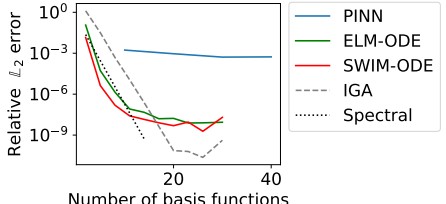

Figure 4: Growth of test error with varying flow velocities $\beta$ for different PDE solvers (14 basis functions) and IGA-FEM (15 basis functions).

Figure 5: Fast exponential decay of test error with the number of neurons in the hidden layer (number of basis functions) for $\beta = 40$.

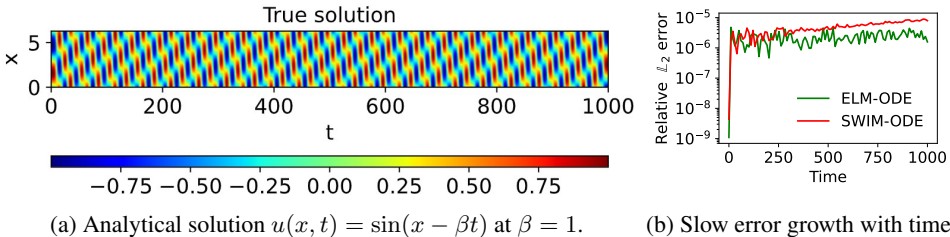

(a) Analytical solution $u(x,t) = \sin(x - \beta t)$ at $\beta = 1$.  (b) Slow error growth with time

Figure 6: Advection equation: Long time simulation at $\beta = 1$.

## 4.2 HIGHER-ORDER DERIVATIVES IN SPACE AND TIME

In this example, the beam equation ($u_{tt} + u_{xxxx} = f$, also see Appendix C.2) with fourth- and second-order derivatives in space and time, respectively, is solved with initial data $u(x, 0) = \sin(x)$ on the spatial domain $\Omega = [0, \pi]$. The force function and the analytical solution are taken from Kapoor et al. (2023). Table 2 shows that ELM-ODE is more than five orders of magnitude faster and more accurate than PINNs.

## 4.3 NON-LINEARITY AND COMPLICATED DOMAIN GEOMETRY

In this example, we demonstrate the efficiency and superior accuracy (by 4 orders of magnitude) of SWIM-ODE in solving a non-linear diffusion equation on a complicated spatial domain compared to PINNs (see Figure 12 for the geometry and Table 2 for results). SWIM-ODE with less than 500 basis functions is three orders of magnitude more accurate than the mesh-based FEM with 2000 finite elements (cf Table 10). We keep the grid points used to generate a mesh in the FEM the same as the data points used to solve the re-formulated ODE with our approach. The details concerning the experiments and boundary conditions can be found in Appendix C.3, Appendix B.1.1, respectively.

## 4.4 NON-LINEARITY AND SHOCKS

We demonstrate how sampling weights and biases from a data-dependant distribution can be exploited to handle locally steep gradients in the solution of the non-linear viscous Burgers' equation. In Figure 8, we compare the SWIM-ODE solution to the numerical solution provided by Raissi et al. (2019). Table 19 indicates that ELM-ODE cannot accurately represent the sharp gradient in the domain's center due to the exponentially small probability of having large norms of internal weights. Sampling ELM-ODE weights from a broader uniform distribution increases the probability of having steeper basis functions, as Calabrò et al. (2021) discuss for linear PDEs. However, given enough collocation points in the domain's center, SWIM-ODE can create numerous basic functions with steep gradients, accurately placing them in the domain's center by factoring in the data. To concentrate collocation points near the shock in the domain's center, we resample them two times after a set number of time steps, guided by a probability distribution that leverages the gradient of the approximated solution. At the resampling time $t_r \in [0, T]$, we approximate the probability density $p(x) \sim |\nabla \hat{u}(x, t_r)|$, which we then use to re-sample collocation points at random. While PINNs provide a reasonable error, SWIM-ODE is more accurate by order of magnitude, almost twice as fast as regular PINN, and over ten times faster than causal PINN (See Table 2, Table 19). Please refer to Appendix C.4 for details. We also demonstrate with a snapshot of the Burgers' solution that SWIM basis functions exhibit a rapid exponential decay of error with increasing network width, where Fourier and Chebyshev basis functions used in classical spectral methods suffer from the Gibbs phenomenon Gottlieb & Shu (1997) (See Appendix C.4.1).

## 4.5 HIGH-DIMENSIONALITY

The goal of this example is to highlight our algorithm's ability to solve high-dimensional PDEs efficiently, unlike the vanilla FEM and spectral methods, which suffer from exponential growth in grid points and basis functions as the dimension increases. We demonstrate in Figure 7 that ELM-ODE can accurately solve the heat equation accurately in 3, 5, 7, and 10 dimensions. For the 3-dimensional heat equation, ELM-ODE is around 10000 times more accurate and 100 times faster than PINNs.

Table 2: Summary of the results of the computational experiments (a detailed comparison with more neural PDE solvers in Appendix C). We outperform PINNs trained with backpropagation by 1-5 orders of magnitude in accuracy and up to 4 orders of magnitude in training time. The results are even comparable to the state-of-the-art mesh-based solvers (shown in italics) while retaining all the advantages of mesh-free methods. Note that the number of basis functions differs for all methods and was chosen to optimize the individual results. For SWIM-ODE, the number of basis functions is always much lower than the finite elements used in the mesh-based methods.

| PDE | Method | Training time (s) | Relative $L^2$ error |
|---|---|---|---|
| Advection ($\beta = 40$) | PINN | 30.5 | 6.92e-1 $\pm$ 2.96e-2 |
| | **ELM-ODE (our)** | 2.7 | 3.84e-6 $\pm$ 5.2e-7 |
| | *Mesh-based method (IGA)* | 0.07 | 1.17e-10 |
| Euler-Bernoulli | PINN | 2303.71 | 4.21e-3 $\pm$ 9.56e-4 |
| | **ELM-ODE (our)** | 0.06 | 3.50e-8 $\pm$ 7.79e-9 |
| | *Mesh-based method (IGA)* | 0.94 | 4.21e-7 |
| Burgers | PINN | 275.2 | 3.88e-3 $\pm$ 2.61e-3 |
| | **SWIM-ODE (our)** | 141.5 | 3.33e-4 $\pm$ 4.63e-4 |
| | *Mesh-based method (IGA)* | 13.61 | 2.20e-4 |
| Nonlinear diffusion | PINN | 143.3 | 1.22e-2 $\pm$ 2.38e-4 |
| | **SWIM-ODE (our)** | 423 | 2.00e-6 $\pm$ 1.99e-6 |
| | ELM-ODE (our) | 4.8 | 7.34e-3 $\pm$ 1.8e-3 |
| | *Mesh-based method (FEM)* | 2.71 | 2.68e-3 |
| 10-d heat equation | PINN | 189.6 | 6.06e-4 $\pm$ 1.00e-4 |
| | **ELM-ODE-fast (our)** | 0.65 | 7.18e-4 $\pm$ 3e-4 |
| | **ELM-ODE-accurate (our)** | 168.6 | 2.28e-5 $\pm$ 2.1e-5 |

For the 10-dimensional heat equation, ELM-ODE-fast has a lower width of 500 and matches the accuracy of PINNs but is 300 times faster to train, whereas ELM-ODE-accurate, with a higher width of 4000, is 25 times more accurate than PINNs. Note that if we consider 10 grid points per dimension for the FEM, one would need 10 billion grid points, whereas our approach requires around 3000 basis functions for the 10-dimensional heat equation. We also observe in Figure 7 that the error decays rapidly (roughly exponentially until dimension 5) for ELM-ODE until it plateaus at a certain network width. Moreover, the error is uniformly low in different parts of the domain. Due to the smoothness and lack of steep gradients in the solution of the PDE, ELM-ODE is clearly more suitable for approximating the solution of the chosen PDE and is one to three orders of magnitude more accurate than vanilla SWIM-ODE. Please refer to Appendix C.5 for details. We summarize the utility of our algorithm compared to the classical general-purpose methods in Table 3.

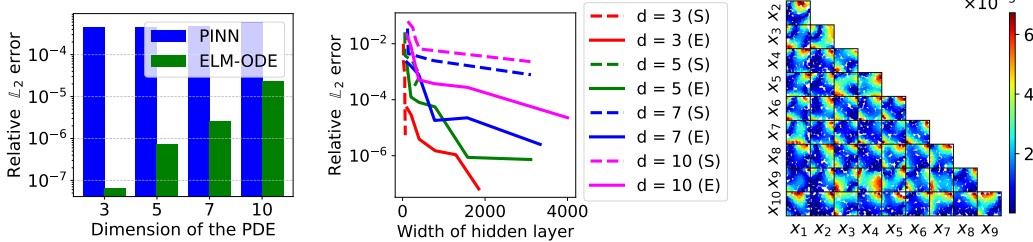

Figure 7: High-dimensional heat equation: (Left) Comparison of test errors for varying dimensions of the PDE, (Middle): Fast decay of test error with the number of neurons in the hidden layer (S: SWIM-ODE), (E: ELM-ODE), (Right): Pointwise test errors of ELM-ODE evaluated at 100 test points each in different 2-dimensional slices (all other dimensions set as the center values of the spatial domain) and time $t = 0.5$.

Table 3: Comparison of our algorithm with classical mesh-based FEM in different problem settings presented in this paper. Our algorithm (SWIM-ODE / ELM-ODE) is fast, accurate, easy to implement, and robust to the different PDE settings (shocks, complicated geometries, high-dimensional PDEs). We use the acronym (CoD) for the Curse of Dimensionality in the following.

| PDE setting | FEM | PINNs | SWIM-ODE / ELM-ODE |
|---|---|---|---|
| Solutions with shocks | ✓ | ✓ | ✓ (SWIM-ODE) |
| Complex domains | Difficult to mesh | Easy | Easy |
| High dimensionality | ✗ (CoD) | ✓ | ✓ |
| Accuracy | High | Low | High |
| Speed | Fast | Training (slow) | Fast |

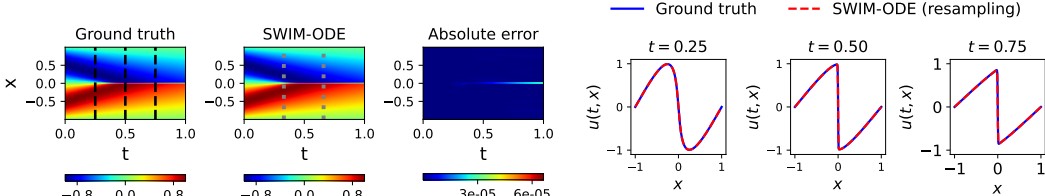

Figure 8: Comparison of SWIM-ODE solution to Burgers' equation and ground truth. Black dashed lines indicate the times at which the solution is compared on the right. Gray dotted lines indicate the times at which the collocation points are re-sampled.

## 5 CONCLUSION

To address the fundamental difficulties stemming from the gradient-based iterative optimization of neural network parameters, we propose a backpropagation-free algorithm for training neural PDE solvers by combining ideas of separation of variables and random sampling of hidden layer parameters.

**Benefits of our method:** Firstly, we demonstrate that our backpropagation-free algorithm for training neural PDE solvers is 2 to $30,000$ times faster and, at the same time, $10$ to $100,000$ times more accurate than the physics-informed neural networks trained with backpropagation for the PDEs considered in this paper. Secondly, by leveraging classical ODE solvers with adaptive time-stepping, we demonstrate that our neural PDE solver can capture high-frequency temporal dynamics and can solve over long time spans, where traditional state-of-the-art neural-PDE solvers fail. Thirdly, our approach reduces the accuracy gap with mesh-based solvers while retaining advantages like mesh-free basis functions, ease of implementation, ability to handle complex domains, and spectral convergence for PDEs with smooth solutions. Finally, we show that our approach can solve high-dimensional PDEs efficiently and accurately, as illustrated by the ten-dimensional heat equation.

**Limitations and future work:** Our approach requires knowledge of the PDE, so grey-box settings and inverse problems, where parts of the PDE must be estimated, are challenging. However, the much faster time-to-solution of our approach should prove very useful for this inverse problem setting. Compared to neural PDE solvers trained with backpropagation, our networks may require more neurons for the same accuracy, leading to higher inference times. Universal approximation properties concerning specific PDE settings and understanding the role of re-sampling network parameters in overcoming the Kolmogorov n-width barrier Peherstorfer (2022) are some of the important theoretical open areas of investigation. Lastly, solving high-dimensional PDEs with complicated solutions is still an interesting avenue to explore further, where traditional numerical methods have many difficulties due to the curse of dimensionality. We hope that our approach can open doors for neural PDE solvers to deal with real-world applications in science and engineering, especially applications where limited accuracy and long training times have been the main reasons for the lack of success.

**Reproducibility Statement:** The code to reproduce the experiments from the paper and an up-to-date code base can be found in the supplemental material and will be published as an open-source repository upon acceptance. We also run with different seeds to compute mean and standard deviations of the results, and store all seeds in the repository to enforce reproducibility. The dataset describing the complicated geometry used in Section 4.3 is included in the supplemental material, and the details to reproduce all experiments can be found in Appendix C.

**Ethics Statement:** Ethical considerations are important for any new machine learning approach because neural networks are generally dual-use. Our approach is based on classical methods from scientific computing, which are well understood. This connection now allows researchers to better understand our neural solvers' behavior, failure modes, and robustness. We believe that the benefits of our approach far outweigh the potential downsides of misuse because a system that is understood better can also be controlled more straightforwardly.

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

APPENDIX

CONTENTS

## A    EXTENDED REVIEW OF RELATED WORK

**Spectral methods for solving PDEs** promise fast convergence with much fewer basis functions. Meuris et al. (2023) present a method to extract hierarchical spatial basis functions from a trained DeepONet and employ it in a spectral method to solve the given PDE. Xia et al. (2023) integrate adaptive techniques into PINN-based PDE solvers to obtain numerical solutions of unbounded domain problems that standard PINNs cannot efficiently approximate. Lange et al. (2021) propose spectral methods that fit linear and nonlinear oscillators to data and facilitate long-term forecasting of temporal signals. Dresdner et al. (2022) demonstrate spectral solvers that provide sub-grid corrections to classical spectral methods to improve their accuracy. Du et al. (2023) use fixed orthogonal bases to learn PDE solutions as a map between spectral coefficients and introduce a training strategy based on spectral loss. These methods differ from ours in problem setting, architecture, and training.

**Neural operator frameworks** (cf. Lu et al. (2021a); Kovachki et al. (2021); Li et al. (2020); Pfaff et al. (2021)) are promising but are typically trained with PDE solutions with different initial conditions, spatial domains (geometries), or parameter settings. Instead, in our setting here, we solve the PDE using given coefficients, domain, and initial conditions without relying on any training data. The ease of implementation, rapid training, and high accuracy of our backpropagation-free approach can be leveraged to generate PDE solution data for training operator networks.

**Mesh-free methods** are typically based on radial basis functions (RBFs, cf. Powell (1992); Chen et al. (2014)) or moving least squares (MLS, cf. Shepard (1968); Lancaster & Salkauskas (1981)). These often do not have user-friendly software or are only applicable in specialized settings (e.g., smoothed particle hydrodynamics, cf. Lucy (1977); Gingold & Monaghan (1977); Shadloo et al. (2016)). Moreover, despite the ease of dealing with complicated geometries, these methods typically suffer from many challenges, such as the choice of kernel, imposing boundary conditions, and convergence issues. These methods are not the focus of this work.

## B    METHODS

### B.1    SWIM-ODE AND ELM-ODE

#### B.1.1    HANDLING BOUNDARY CONDITIONS

Our approaches to satisfying the Dirichlet and periodic boundary conditions are already explained in the main text. Here, we explain how we handle time-dependent Dirichlet boundary conditions and Neumann boundary conditions.

**Time-dependent Dirichlet boundary conditions:** For handling time-dependent Dirichlet boundary conditions ($u(x, t) = g(x, t)$ for $x \in \partial\Omega$), we set $A$ to the identity map and augment the ODE (Equation (4)) with an additional equation,

$$\hat{u}_t(x, t) = g_t(x, t) \text{ for } x \in \partial\Omega \implies C_t(t) = \underbrace{[R(X, C(t)), g_t(X_b, t)]}_{\in \mathbb{R}^{1 \times (N_c + N_b)}} \underbrace{\Phi_A([X, X_b])^+}_{\in \mathbb{R}^{(N_c + N_b) \times (M_b + 1)}},$$

In the example in Section 4.3, we know the solution on the boundary at all time points, which is continuously differentiable. If the solution on the boundary points is not available at all time points, one can interpolate and approximate the derivative of the solution on the boundary.

**Neumann boundary conditions:** For simple spatial domains, one can choose appropriate outer basis functions as described in Appendix B.1.1 that inherently satisfy the Neumann boundary conditions. For instance, for zero Neumann boundary conditions on a one-dimensional domain, one can choose outer basis functions consisting of cosines of different frequencies scaled to the domain (function value is 1 at the boundaries) so that their spatial derivatives, which are the sine functions, are zero on the boundary points.

On complicated domain geometries, to satisfy Neumann boundary conditions ($\nabla u(x, t) \cdot \hat{n}(x) = 0$ for $x \in \partial\Omega$), we set $A$ to the identity map and augment the ODE (Equation (4)) with an additional equation for the boundary points and solve

$$C_t(t) = \underbrace{[R(X, C(t)), 0]}_{\in \mathbb{R}^{1 \times (N_c + N_b)}} \underbrace{[\Phi_A(X), \nabla\Phi_A(X_b)[\hat{n}(X_b)]^\top]^+}_{\in \mathbb{R}^{(N_c + N_b) \times (M_b + 1)}}.$$

### B.1.2 DERIVATION OF SPECIFIC FORM OF ODEs

We use the notation described in Section 3 of the manuscript. We first discuss how to compute different spatial and temporal derivative terms appearing in the PDEs described in this manuscript using the neural network ansatz. We then use these expressions to reformulate the PDEs described in this manuscript as corresponding ODEs. We consider neural networks in the most general setting by considering the outer basis functions and the SVD layer (cf Algorithm 1).

Computing spatial and temporal derivatives of the neural network solution

**Computing spatial derivatives:** We compute the first-order spatial derivative as

$$
\begin{aligned}
\hat{u}_x(x,t) &= C(t)[\Phi_{A_r}]_x(x) \\
&= C(t)[A_r W \odot \tilde{\sigma}_x(x), 0] \in \mathbb{R}^{1 \times d},
\end{aligned}
\tag{6}
$$

where $\odot$ is the Hadamard product,

$$
\tilde{\sigma}_x(x) := [\sigma_z(z)|_{z=Wx^\top+b}, \sigma_z(z)|_{z=Wx^\top+b}, \dots, \sigma_z(z)|_{z=Wx^\top+b}] \in \mathbb{R}^{M_s \times d},
\tag{7}
$$

with $\sigma_z(z) \in \mathbb{R}^{M_s}$ and $\sigma_z$ is the first derivative of the $tanh$ activation function.

Similarly, we compute the second- and fourth-order spatial derivatives as:

$$
\begin{aligned}
\hat{u}_{xx}(x,t) &= C(t)[\Phi_{A_r}]_{xx}(x) \\
&= C(t)[A_r W \odot W \odot \tilde{\sigma}_{xx}(x), 0] \in \mathbb{R}^{1 \times d},
\end{aligned}
\tag{8}
$$

where $\tilde{\sigma}_{xx}(x)$ is defined equivalently as $\tilde{\sigma}_x(x)$ but with $\sigma_{xx}$ being the second-order spatial derivative of the $tanh$ activation function.

$$
\begin{aligned}
\Delta \hat{u}(x,t) &= C(t)[\Phi_{A_r}]_{xx}(x)\mathbb{1}, \quad \text{where, } \mathbb{1} \in \mathbb{R}^{d \times 1} \\
&= C(t)[A_r W \odot W \odot \tilde{\sigma}_{xx}(x), 0]\mathbb{1} \in \mathbb{R}^{1 \times 1},
\end{aligned}
\tag{9}
$$

Finally,

$$
\begin{aligned}
\hat{u}_{xxxx}(x,t) &= C(t)[\Phi_{A_r}]_{xxxx}(x) \\
&= C(t)[A_r W \odot W \odot W \odot W \odot \tilde{\sigma}_{xxxx}(x), 0] \in \mathbb{R}^{1 \times d},
\end{aligned}
\tag{10}
$$

where $\sigma_{zzzz}$ is the fourth-order spatial derivative of the $tanh$ activation function.

**Computing time derivatives:**

We compute the first-order time derivative as

$$
\hat{u}_t(x,t) = C_t(t)[\Phi_{A_r}](x).
\tag{11}
$$

Similarly, we compute the second-order time derivative as

$$
\hat{u}_{tt}(x,t) = C_{tt}(t)[\Phi_{A_r}](x).
\tag{12}
$$

Reformulating PDEs as ODEs  We now reformulate the PDEs as ODEs using the space and time derivatives derived in Appendix B.1.2. We denote the pseudo-inverse by $\cdot^+$.

**Advection equation:**  The one-dimensional advection equation is

$$
u_t(x,t) + \beta u_x(x,t) = 0,
$$

where $\beta$ is a scalar. Approximating the solution with neural network ansatz (Equation (3)) and substituting Equation (11) and Equation (6) in the advection equation, we get,

$$
\begin{aligned}
C_t(t)[\Phi_{A_r}(X)] &= -\beta C(t)[\Phi_{A_r}(X)]_x \\
C_t(t) &= -\beta C(t)[\Phi_{A_r}(X)]_x[\Phi_{A_r}(X)]^+.
\end{aligned}
$$

The initial condition is given by

$$
C(0) = u(X,0)^\top[\Phi_{A_r}(X)]^+.
$$

**Burgers' equation:** The one-dimensional Burgers' PDE we consider is

$$u_t + uu_x - \alpha u_{xx} = 0,$$

where $\alpha$ is a scalar. Approximating the solution with neural network ansatz (Equation (3)) and substituting Equation (11), Equation (6) and Equation (8) in the Burgers equation, we get,

$$C_t(t)\Phi_{A_r}(X) = -\left(C(t)\Phi_{A_r}(X) \odot C(t)[\Phi_{A_r}]_x(X)\right) + \alpha\left(C(t)[\Phi_{A_r}]_{xx}(X)\right)$$

$$C_t(t) = -\left(C(t)\Phi_{A_r}(X) \odot C(t)[\Phi_{A_r}]_x(X) + \alpha\left(C(t)[\Phi_{A_r}]_{xx}(X)\right)\right)[\Phi_{A_r}(X)]^+$$

Note that the non-linearity is transferred to the right-hand side of the ODE. The initial condition is given by

$$C(0) = u(X,0)^\top \Phi(X)^+.$$

**Euler-Bernoulli equation:** The Euler-Bernoulli PDE considered in this manuscript is

$$u_{tt} + u_{xxxx} = f(x,t).$$

Approximating the solution with neural network ansatz (Equation (3)) and substituting Equation (10) and Equation (12) in the Euler-Bernoulli equation, we get,

$$C_{tt}(t)\Phi(X) = f(X,t)^\top - C(t)\Phi_{xxxx}(X)$$

We re-write this second-order ODE as a combination of first-order ODEs given by

$$C_t(t) = D(t),$$

$$D_t(t)\Phi(X) = f(X,t)^\top - C(t)\Phi_{xxxx}(X).$$

We then reformulate the ODEs as

$$\begin{pmatrix} C_t(t) & D_t(t) \end{pmatrix} = \begin{pmatrix} C(t) & D(t) \end{pmatrix}\begin{pmatrix} 0 & -\Phi(X)_{xxxx}\Phi(X)^+ \\ \mathbb{1} & 0 \end{pmatrix} + \begin{pmatrix} 0 & \mathbb{1} \end{pmatrix}[f(X,t)]^\top \Phi(X)^+.$$

The initial condition is given by

$$C(0) = u(X,0)^\top \Phi(X)^+,$$

$$D(0) = u_t(X,0)^\top \Phi(X)^+.$$

**Nonlinear diffusion equation:** The two-dimensional nonlinear diffusion equation we consider is

$$u_t - u\Delta u = f(x,t), \quad x \in \Omega \subset \mathbb{R}^2, \quad t \in [0,1]. \tag{13}$$

Approximating the solution with neural network ansatz (Equation (3)), substituting Equation (11), and Equation (8) in the nonlinear diffusion equation, we get,

$$C_t(t)\Phi(X) = (C(t)\Phi(X) \odot [C(t)\Phi_{xx}(X)]\mathbb{1}) + [f(X,t)]^\top$$

$$C_t(t) = \left(C(t)\Phi(X) \odot [C(t)\Phi_{xx}(X)]\mathbb{1} + [f(X,t)]^\top\right)\Phi(X)^+$$

Note that the non-linearity is transferred to the right-hand side of the ODE. The initial condition is given by

$$C(0) = u(X,0)^\top \Phi(X)^+$$

**High-dimensional diffusion equation:** The d-dimensional diffusion equation we consider is

$$u_t - \Delta u = f(x,t), \quad x \in \Omega \subset \mathbb{R}^d, \quad t \in [0,1]. \tag{14}$$

Approximating the solution with neural network ansatz (Equation (3)), substituting Equation (11), and Equation (8) in the diffusion equation, we get,

$$C_t(t)\Phi(X) = [C(t)\Phi_{xx}(X)]\mathbb{1} + [f(X,t)]^\top$$

$$C_t(t) = \left([C(t)\Phi_{xx}(X)]\mathbb{1} + [f(X,t)]^\top\right)\Phi(X)^+$$

The initial condition is given by

$$C(0) = u(X,0)^\top \Phi(X)^+$$

**Note on ODE solvers and interpolation in time:** We use the `solve_ivp` routine of the SciPy package Virtanen et al. (2020). One can pass test points in time as an argument to the method `solve_ivp`. One can optionally set the parameter `dense_output` to true, which means that the output of the ODE is a function handle that can be evaluated by interpolation at any time point $t \in \Omega$. The method specified dictates the interpolation order. RK23 uses a cubic Hermite polynomial, while DOPRI85 uses a seventh-order polynomial.

### B.1.3 EXTENDED DISCUSSION OF OUR METHOD

**Kolmogorov n-width barrier:** Without resampling the internal network parameters, our method faces the Kolmogorov n-width barrier Peherstorfer (2022); Du & Zaki (2021); Berman & Peherstorfer (2024); Kast & Hesthaven (2024) because our basis functions are not time-dependent. However, resampling basis functions at certain time points of the SWIM-ODE (as done in the Burgers' equation in Section 4.4) results in a solution- and time-dependent basis approximation of the solution manifold and, thus, in theory, can break the barrier.

PINNs can theoretically break the Kolmogorov n-width barrier as time is treated as an extra spatial dimension, and internal network parameters are time-dependent. However, for PINNs, the optimization issues pose much more severe challenges even on very simple PDEs and in low dimensions (Krishnapriyan et al., 2021; Wang et al., 2021; 2022). So even though our vanilla SWIM-ODE/ELM-ODE approach (without periodically resampling hidden layer weights) faces the Kolmogorov n-width barrier, we outperform PINNs, typically by several orders of accuracy and time in practice.

**Influence of random sampling on the method:** Similar to the question of how PINNs trained with Adam/SGD perform based on their random network initialization, understanding the influence of weights on the output is a challenge. There are two main differences between (stochastic) gradient optimization and our setting. First, after fixing the internal weights, we use regularized least-squares (not a stochastic method) to fit the initial condition. Second, we do not use a stochastic method to solve over time. Therefore, even though PINNs can adapt their random initialization over the gradient-based optimization, precisely that optimization also adds stochasticity. If the number of neurons for the model increases, the randomness in our case decreases because the regularized least-squares fit to the initial condition (which converges to a single solution in the limit of many neurons), while stochastic gradient descent will only converge to a distribution (because of mini-batch optimization). This has been observed for the supervised learning problems in Bolager et al. (2023), particularly in the transfer learning experiments. In Table 2, we observe that our model's performance is often orders of magnitude better, and the variance is on the same scale as the magnitude.

**Comparison between SWIM-ODE and ELM-ODE:** One of the main factors influencing the performance of SWIM-ODE and ELM-ODE is the underlying solution of the PDE.

We explain, with an example of the Burgers' equation, how the SWIM sampling can be leveraged when the solution has steep gradients, as one can sample localized basis functions in the part of the domain where the solution has steep gradients. For ELM, the probability of sampling steep basis functions with the vanilla ELM is lower, as illustrated in the Figure 2 of the paper. Even if one uses a different distribution to sample the network parameters such that more basis functions with steep gradients are sampled, placing the basis functions at appropriate spatial locations is another challenge. With ELM, one cannot resample or choose basis functions using data as it is data-agnostic. Thus, especially if the solution has localized steep gradients, the performance of ELM is typically worse compared to SWIM. We additionally demonstrate with a snapshot of the Burgers' solution that SWIM basis functions exhibit a rapid exponential decay of error with increasing network width, where ELM, Fourier, and Chebyshev basis functions used in classical spectral methods suffer from the Gibbs phenomenon (see Appendix C.4.1) and lead to poor scaling and accuracy (see Figure 15, Figure 16).

If the underlying solution is sufficiently smooth and does not have steep gradients anywhere in the domain, ELM typically performs very well, as seen in the example with the Advection equation (see Section 4.1), Euler Bernoulli equation (see Section 4.2), and the effect is most apparent in the newly added example of high-dimensional PDEs (Section 4.5), where ELM-ODE performs much better than SWIM-ODE as shown in Table 22. While we just use the vanilla SWIM algorithm in the presented results, one can easily adapt the algorithm and, after sampling the network parameters with SWIM, multiply the basis functions with a tunable scaling factor before applying the non-linearity to sample many more basis functions with shallow slopes.

Thus, the choice between the two strategies is particularly governed by the underlying solution of the PDE. Apart from the favorable cases for each method mentioned above, both methods have comparable performance and typically outperform PINNs by several orders of magnitude in speed and time. Thus, the rapid training of our approach could be leveraged to try out both approaches if one has no information about what the solution of the PDE could look like.

**A discussion on "data-driven" and "data-agnostic" sampling:** We emphasize what we mean by data-driven sampling: our data are random pairs of collocation points, but we do not have access to the solution function values (because, in the beginning, we have not solved the PDE yet). Thus, even though we do not have access to the true solution of the PDE, we call this "data-driven" sampling because we create the parameters of our basis functions (neurons) so that they are centered strictly within the domain. We achieve this by using data points sampled in the domain, thereby considering the geometry and bounds of the spatial domain. Note that with data-agnostic sampling in ELM, the neurons can easily be centered outside the spatial domain because weights and biases are chosen without considering any information about the geometry and bounds of the spatial domain. To summarize, though our algorithm proposes "data-driven" sampling, we do not start with time-series data and instead work in a self-supervised setting.

### B.2 Physics-Informed Neural Networks

This work employs two prominent variants of physics-informed machine learning to compare the results obtained by the proposed methods. In particular, physics-informed neural network (PINN) Raissi et al. (2019) and causality-respecting physics-informed neural network (causal PINN) Wang et al. (2024b) are utilized to compare the obtained approximations.

Vanilla PINNs are feedforward deep neural networks designed to simulate PDEs by incorporating physical laws into the learning process. The architecture of a vanilla PINN includes a deep neural network that maps inputs (e.g., space and time coordinates) to outputs (e.g., physical quantities of interest) and is trained to minimize a loss function that combines both data and physics-based errors. The data term ensures that the neural network fits the provided data points, while the physics term enforces the PDE constraints with automatic differentiation. Hence, the loss function for PINN could be defined by

$$L(\mu) = \lambda_1 L_{\text{PDE}}(\mu) + \lambda_2 L_{\text{Data}}(\mu). \tag{15}$$

Here, $\mu$ represents the trainable network parameters. Considering the generic nonlinear PDE defined by equation 1 with well-posed boundary and initial conditions equation 2, the individual loss terms weighted by the hyperparameters $\lambda_i$, $i = 1, 2$, are defined by

$$L_{\text{PDE}}(\mu) = \frac{1}{N_{\text{int}}} \sum_{n=1}^{N_{\text{int}}} ||u_t^*(x^{(n)}, t^{(n)}) + Lu(x^{(n)}, t^{(n)}) + \lambda N(u^*)(x^{(n)}, t^{(n)}) - f(x^{(n)})||^p. \tag{16}$$

The data loss term considering the initial and boundary conditions is defined by

$$L_{\text{Data}}(\mu) = \frac{1}{N_{\text{i}}} \sum_{n=1}^{N_{\text{i}}} ||Bu^*(x^{(n)}, t^{(n)}) - g(x^{(n)})||^p. \tag{17}$$

Here, $N$ is the total number of training points, which is the sum of interior training points ($N_{\text{int}}$) and initial or boundary training points ($N$). The neural network predicted solution of $u$ at a point in computational domain, $(x^{(n)}, t^{(n)})$ is denoted by $u^*(x^{(n)}, t^{(n)})$. The experiments are trained with $L^2$-norm, implying $p = 2$. The main goal is to minimize equation 15 and determine the optimal parameters ($\mu$). These parameters, once optimized, are employed to predict the solution of the PDE within the computational domain.

The second physics-informed method employed is Causal PINNs, which modifies the PINN loss function to explicitly adhere to the temporal causality inherent in time-dependent PDEs. In vanilla PINNs, the loss function does not prioritize resolving the solution at lower times before higher times, leading to inaccuracies, especially in time-dependent problems. Causal PINNs address this by introducing a weighting factor for the loss at each time step, which depends on the accumulated loss from previous time steps. The resulting loss function ensures that the network prioritizes learning the solution accurately at earlier times before focusing on later times, thus maintaining the causal structure of the physical problem being solved. The causal PDE loss term is defined by

$$L_{\text{PDE}}(\mu) = \sum_{i=1}^{N_{\text{t}}} w_i L_{\text{PDE}}(t_i, \mu), \tag{18}$$

$$w_1 = 1, \quad w_i = e^{-\epsilon \sum_{k=1}^{i-1} L_{\text{PDE}}(t_k, \mu)}, \quad i = 2, 3, \ldots N_{\text{t}}.$$

$N_t$ represents the number of time steps into which the computational domain is divided. The causality hyperparameter $\epsilon$ regulates the steepness of the weights and is incorporated in the loss function similar to Kapoor et al. (2024b). This modification introduces a weighting factor, $w_i$, for the loss at each time level $t_i$, with $w_i$ being dependent on the cumulative PDE loss up to time $t_i$. The network prioritizes a fully resolved solution at earlier time levels by exponentiating the negative of this accumulated loss. Consequently, the modified loss function for causal PINN is expressed as

$$L_{\text{PDE}}(\mu) = \frac{1}{N_t}\left[w_1 L_{\text{PDE}}(t_1, \mu) + \sum_{i=2}^{N_t} e^{-\epsilon \sum_{k=1}^{i-1} L_{\text{PDE}}(t_k, \mu)} L_{\text{PDE}}(t_i, \mu)\right]. \tag{19}$$

### B.3 IGA-FEM

First introduced in Hughes et al. (2005), Isogeometric analysis (IGA) is a numerical method developed to unify the fields of computer-aided design (CAD) and finite element analysis (FEA). The key idea is to represent the solution space for the numerical analysis using the same functions that define the geometry in CAD (cf. Cottrell et al. (2009)), which include the B-Splines and Non-Uniform Rational B-Splines (NURBS) (cf. Piegl & Tiller (1997)).

In this paper, we use B-Splines as the basis functions. The B-Splines are defined using the Cox-de Boor recursion formula (cf. COX (1972); de Boor (1972)), i.e.,

$$N_{i,0}(\xi) = \begin{cases} 1 & \xi_i \leq \xi < \xi_{i+1} \\ 0 & \text{otherwise}, \end{cases}$$

$$N_{i,p}(\xi) = \frac{\xi - \xi_i}{\xi_{i+p} - \xi_i} N_{i,p-1}(\xi) + \frac{\xi_{i+p+1} - \xi}{\xi_{i+p+1} - \xi_{i+1}} N_{i+1,p-1}(\xi),$$

where $\xi_i$ is the $i$th knot, and $p$ is the polynomial degree. The vector $\Xi = [\xi_1, \xi_2, \ldots, \xi_{n+p+1}]$ is the knot vector, where $n$ is the number of B-Splines. By specifing the knot vector, we define the basis functions we use to solve the PDEs. We use an uniform open knot vector, where the first and last knots have multiplicity $p + 1$, the inner knots have no multiplicity, and all knots that have different values are uniformly distributed. We refer to the knots with different values as "nodes". The intervals between two successive nodes are knot spans, which can be viewed as "elements". The elements form a "patch". A domain can be partitioned into subdomains and each is represented by a patch. In our work, we use a single patch to represent the entire 1D domain. Figure 9a shows an example of such a patch, where the B-Splines are $C^p$-continuous within the knot spans and $C^{p-1}$ continuous at the inner knots. In order to address the boundary conditions, we adapt the B-Splines as shown in Figure 9b Figure 9c, so that the boundary conditions are directly built into the solution space.

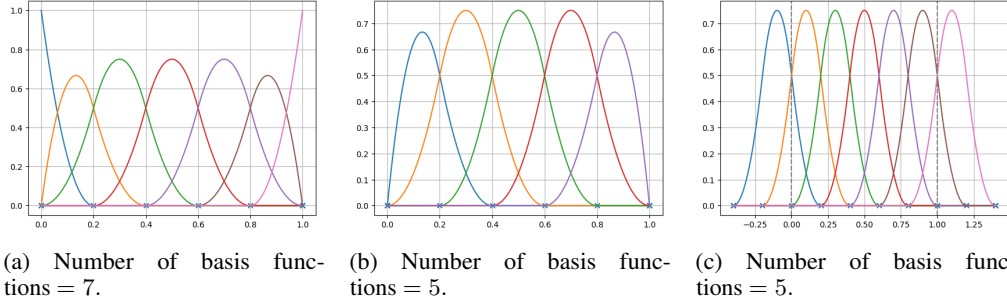

(a) Number of basis functions = 7.

(b) Number of basis functions = 5.

(c) Number of basis functions = 5.

Figure 9: Examples of B-Splines representing the 1D domain $[0, 1]$. Number of nodes = 6 and degree of polynomials = 2. **Left:** The original B-Splines. **Middle:** Adapted B-Splines to satisfy the Dirichlet boundary condition. **Right:** Adapted B-Splines to satisfy the periodic boundary condition. Note that the first (blue) spline is identical to the second last (brown) one, and the second (orange) spline is identical to the last (pink) one, as they share the same coefficient. The gray dashed lines indicate where the domain starts and ends.

In the following, we refer to the adapted B-Splines as basis functions $\phi_k(x)$. Thus, the solutions of PDEs are approximated by

$$u(x,t) = \sum_{k=1}^{K} c_k(t)\phi_k(x).$$

We solve the PDEs in the weak formulation. For the linear advection equation equation 21, the weak form of the equation is

$$\sum_{k=1}^{K} c_k'(t) \int_X \phi_k(x)v(x)dx + \beta \sum_{k=1}^{K} c_k(t) \int_X \phi_k'(x)v(x)dx = 0, \tag{20}$$

where $v(x)$ are the test functions. The test functions are chosen to be the same as the basis functions. The integral of the functions is computed using Gaussian quadrature. Then we solve the linear Ordinary differential equation (ODE)

$$\mathbf{M}\dot{\mathbf{c}} + \mathbf{K}\mathbf{c} = \mathbf{0},$$

where matrix $\mathbf{M}$ and matrix $\mathbf{K}$ contain the integral of the B-Splines and their derivatives, and the coefficient $\beta$, which are given. We solve the Euler-Bernoulli equation equation 23 and the Burgers' equation equation 27 in a similar way. The boundary condition for the Euler-Bernoulli equation is in addition weakly imposed, as is done in Prudhomme et al. (2001).

### B.3.1 DOLFINx

DOLFINx (cf. Baratta et al. (2023)), which is part of the FEniCS project, is a C++ and Python library used for solving PDEs with the finite element method (FEM). It provides tools for defining complex geometries, formulating variational problems, and solving them efficiently on distributed architectures. In this paper, we use `DOLFINx` 0.8.0 to solve the nonlinear diffusion equation equation 24. The programming language is Python. The runtime of the FEM experiment in Table 15 was measured on a Lenovo ThinkPad T14s laptop equipped with 12th Gen Intel® Core™ i7-1255U (12 cores) and 16GB of RAM. In addition, the software Gmsh (cf. Geuzaine & Remacle (2009)) is used to generate mesh for the complex geometry for this experiment, as shown in Figure 12a.

## C  NUMERICAL EXPERIMENTS

**Code repository:** The code to reproduce the experiments from the paper and an up-to-date code base can be found (with MIT Licence) in the supplemental material (open-source and publicly available after acceptance).

**Hardware details:** The computational experiments were performed with: `Ubuntu` 20.04.6 LTS, `NVIDIA` driver 515.105.01 and i7 CPU.

**Metrics for computing errors:** Let $d$ be the dimension of space and $\Omega \times [0,T] \subset \mathbb{R}^d \times \mathbb{R}$ be the spatio-temporal domain. Given $N$ points in a test set $X_{\text{test}}$, the error metrics we use to compare numerical results are Root Mean Squared Error (RMSE) and relative $L^2$ error given by

$$\text{RMSE} := \sqrt{\frac{\sum_{x \in X}(u_{true}(x) - u_{pred}(x))^2}{N}}, \text{ and } L^2_{\text{relative}} := \frac{\sqrt{\sum_{x \in X}(u_{true}(x) - u_{pred}(x))^2}}{\sqrt{\sum_{x \in X}(u_{true}(x))^2}}.$$

The mean and standard deviation of the RMSE and $L^2_{\text{relative}}$ are computed with 3 seeds in all the computational experiments.

### C.1  LINEAR ADVECTION EQUATION

**Problem setup:** The advection equation models the propagation of a quantity at a speed $\beta$ without altering the shape. We solve the linear advection equation with periodic boundary conditions described by

$$u_t(x,t) + \beta u_x(x,t) = 0, \quad \text{for} \quad x \in [0, 2\pi], t \in [0,1], \tag{21}$$

$$u(x,0) = \sin(x). \tag{22}$$

We describe additional details in solving the advection equation with various neural PDE solvers in Table 4. The hidden layer weights for ELM and ELM-ODE are sampled from the Gaussian distribution and biases from a uniform distribution in $[-4, 4]$.

**Ablation studies**: In addition, hyperparameter optimization for PINN for the case of $\beta = 10$ was carried out, varying the number of neurons and interior points. The results for the hyperparameter optimization are detailed in Table 5, Table 6. For SWIM and ELM, we use 1000 interior points for $\beta \in \{10^{-2}, 10^{-1}, 1, 10\}$, and we use 8000 interior points for $\beta \in \{40, 100\}$. The ablation study for the width of the network for SWIM-ODE and ELM-ODE is already presented in the main text in Figure 5. We do not perform ablation studies for the SVD layer in this case, as we do not need to use the SVD layer. Since the width is already quite low for optimal parameters, the SVD layer retains the width.

**Comparison of results:**

Figure 10 shows the absolute errors obtained with the SWIM-ODE, ELM-ODE, PINN, Causal PINN, and IGA methods.

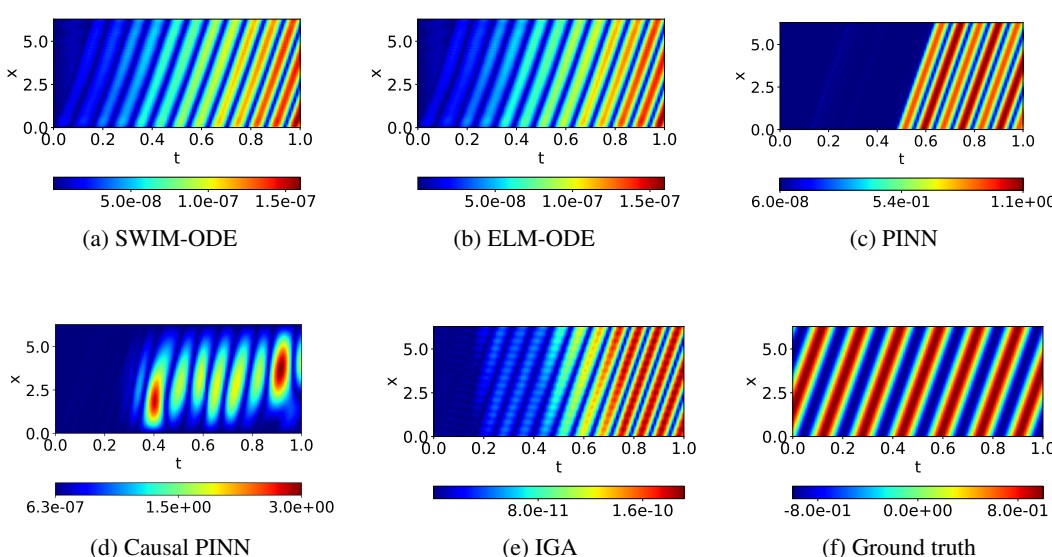

Figure 10: Advection equation: absolute error plots and ground truth

## C.2 EULER-BERNOULLI PDE

**Problem Setup:** This is a time-dependent PDE given by

$$u_{tt} + u_{xxxx} = f(x, t) \quad x \in [0, \pi], t \in [0, 1] \tag{23}$$

where $f(x, t) = (1 - 16\pi^2) \sin(x) \cos(4\pi t)$, with initial and boundary conditions

$$u(x, 0) = \sin(x), \quad u_t(x, 0) = 0$$

$$u(0, t) = u(\pi, t) = u_{xx}(0, t) = u_{xx}(\pi, t) = 0.$$

It models a simply supported beam with varying transverse force. We describe additional details in solving the Euler-Bernoulli with various neural PDE solvers in Table 8. The hidden layer weights for ELM-ODE are sampled from the Gaussian distribution and biases from a uniform distribution in $[-2, 2]$.

**Comparison of results:** Figure 11 shows the absolute errors obtained with the SWIM-ODE, ELM-ODE, PINN, and IGA methods, and Table 9 shows the summary of results for Euler Bernoulli beam equation for different methods.

Table 4: Advection equation: Network hyperparameters used for $\beta \in \{10^{-2}, 10^{-1}, 1, 10, 40, 100\}$ (optimal hyper-parameters in bold)

| | Parameter | Value |
|---|---|---|
| SWIM-ODE, ELM-ODE | Number of hidden layers | 2 |
| | Hidden layer width | $[140, \mathbf{380}, 560]$ |
| | Activation | tanh |
| | $L^2$-regularization | $[10^{-8}, \mathbf{10^{-10}}, 10^{-12}]$ |
| | Loss | mean-squared error |
| SWIM, ELM | Number of hidden layers | 2 |
| | SVD cutoff | $10^{-12}$ |
| | Hidden layer width | $[140, \mathbf{380}, 560]$ |
| | Activation | tanh |
| | $L^2$-regularization | $[10^{-8}, \mathbf{10^{-10}}, 10^{-12}]$ |
| | Loss | mean-squared error |
| | # Initial and boundary points | 400 |
| | # Interior points | $[\mathbf{1000}, \mathbf{8000}]$ |
| IGA | Number of nodes | 16 |
| | Degree of polynomials | 8 |
| | Number of basis functions | 15 |
| PINN | Number of hidden layers | 4 |
| | Layer width | $[10, 20, \mathbf{30}, 40]$ |
| | Activation | tanh |
| | Optimizer | LBFGS |
| | Epochs | 5000 |
| | Loss | mean-squared error |
| | Learning rate | 0.1 |
| | Batch size | 200 |
| | Parameter initialization | Xavier (Glorot & Bengio, 2010) |
| | Loss weights, $\lambda_1, \lambda_2$ | 1, 1 |
| | # Interior points | $[500, 1000, 1500, \mathbf{2000}]$ |
| | # Initial and boundary points | 600 |
| Causal PINN | Number of hidden layers | 4 |
| | Layer width | 30 |
| | Activation | tanh |
| | Optimizer | ADAM followed by LBFGS |
| | ADAM Epochs | 2000 |
| | LBFGS Epochs | 5000 |
| | Loss | mean-squared error |
| | Learning rate | 0.1 |
| | Batch size | 2000 |
| | Parameter initialization | Xavier (Glorot & Bengio, 2010) |
| | Loss weights, $\lambda_1, \lambda_2$ | 1, 1 |
| | # Interior points | 40000 |
| | # Initial and boundary points | 6000 |
| | Causality parameter, $\epsilon$ | 10 |

## C.3 NONLINEAR DIFFUSION EQUATION

**Problem Setup:** We solve a two-dimensional nonlinear diffusion equation

$$u_t - u\Delta u = f(x, y, t), \quad (x, y) \in \Omega, \quad t \in [0, 1], \tag{24}$$

$$f(x, y, t) = 5e^{-t}\sin(\pi x)y^{-3}\left(-1 + e^{-t}\sin(\pi x)y^{-5}\left(-12 + \pi^2 y^2\right)\right) \tag{25}$$

on a complicated geometry inspired by a tree-like pattern occurring during the controlled shaping of fluids (Islam & Gandhi, 2017). The initial condition and time-dependent Dirichlet boundary

Table 5: Advection equation: Ablation study for PINN with respect to the network width for $\beta = 10$. The mean is computed over 3 seeds.

| Layer width | Training time (s) | RMSE | Relative $L^2$ error |
|---|---|---|---|
| 10 | **24.47 ± 0.19** | 1.24e-3 ± 2.38e-4 | 1.76e-3 ± 3.37e-4 |
| 20 | 27.46 ± 0.08 | 6.52e-4 ± 2.59e-4 | 9.22e-4 ± 3.66e-4 |
| 30 | 30.43 ± 0.50 | **3.69e-4 ± 4.33e-5** | **5.23e-4 ± 6.13e-5** |
| 40 | 33.64 ± 0.41 | 3.86e-4 ± 9.37e-5 | 5.46e-4 ± 1.32e-4 |

Table 6: Advection equation: hyperparameter optimization for PINN varying the number of interior points for $\beta = 10$. The mean is computed over 3 seeds.

| Interior points | Training time (s) | RMSE | Relative $L^2$ error |
|---|---|---|---|
| 500 | **25.76 ± 0.29** | 4.10e-4 ± 7.20e-5 | 5.80e-4 ± 1.01e-4 |
| 1000 | 27.44 ± 0.25 | 3.72e-4 ± 4.06e-5 | 5.27e-4 ± 5.74e-5 |
| 1500 | 29.61 ± 0.16 | 5.68e-4 ± 1.97e-4 | 8.03e-4 ± 2.79e-4 |
| 2000 | 30.43 ± 0.50 | **3.69e-4 ± 4.33e-5** | **5.23e-4 ± 6.13e-5** |

Table 7: Advection equation: Summary of results for $\beta = 40$.

| Method | Training time (s) | RMSE | Relative $L^2$ error | architecture |
|---|---|---|---|---|
| SWIM | 66.30 | 6.81 ± 0.26 | 9.63 ± 0.38 | (2, 4000, 1) |
| ELM | 59.05 | 3.78 ± 0.21 | 5.35 ± 0.29 | (2, 4000, 1) |
| Causal PINN | 357.63 ± 3.11 | 2.07 ± 0.87 | 2.92 ± 1.23 | (2, 4 × 30, 1) |
| PINN | 30.56 ± 0.27 | 4.89e-1 ± 2.09e-2 | 6.92e-1 ± 2.96e-2 | (2, 4 × 30, 1) |
| SWIM-ODE (our) | 2.72 | 5.04 e-6 ± 1.45e-6 | 7.13e-6 ± 2.05 e-6 | (1, 350, 15, 1) |
| **ELM-ODE (our)** | **2.71** | **2.73e-6 ± 3.68e-7** | **3.84e-6 ± 5.2e-7** | (1, 350, 15, 1) |
| *IGA-FEM* | 0.07 | 8.24e-11 | 1.17e-10 | 15 |

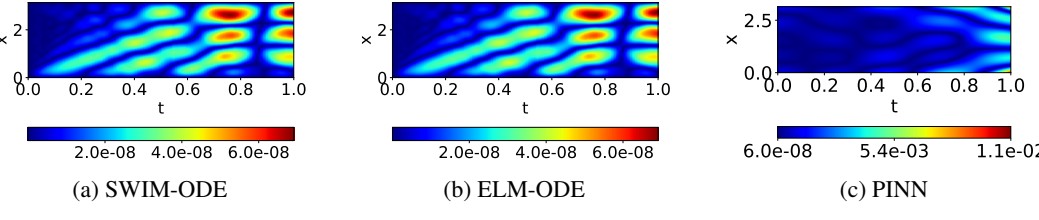

(a) SWIM-ODE  (b) ELM-ODE  (c) PINN

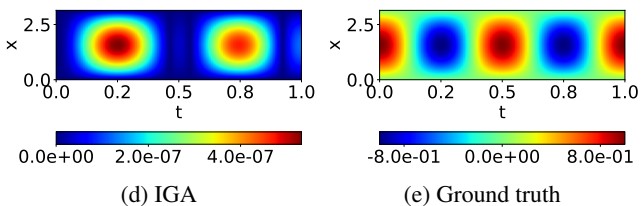

(d) IGA  (e) Ground truth

Figure 11: Euler-Bernoulli equation: absolute error plots and ground truth

conditions are obtained from the constructed solution of the PDE

$$u(x, y, t) = 5e^{-t} \sin(\pi x) y^{-3}, \quad (x.y) \in \Omega, \quad t \in [0, 1]. \tag{26}$$

Table 8: Euler-Bernoulli equation: Summary of all hyperparameters.

|  | Parameter | Value |
|---|---|---|
| SWIM-ODE and ELM-ODE | Number of hidden layers | 2 |
|  | Hidden layer width | 200 |
|  | Outer layer width | 10 |
|  | Activation | tanh |
|  | $L^2$-regularization | $10^{-12}$ |
|  | Loss | mean-squared error |
| IGA | Number of nodes | 27 |
|  | Degree of polynomials | 9 |
|  | Number of basis functions | 33 |
| PINN | Number of hidden layers | 4 |
|  | Layer width | 20 |
|  | Activation | tanh |
|  | Optimizer | LBFGS |
|  | Epochs | 15000 |
|  | Loss | mean-squared error |
|  | Learning rate | 0.1 |
|  | Batch size | 2000 |
|  | Parameter initialization | Xavier (Glorot & Bengio, 2010) |
|  | Loss weights, $\lambda_1, \lambda_2$ | 0.1, 1 |
|  | # Interior points | 10000 |
|  | # Initial and boundary points | 6000 |

Table 9: Euler-Bernoulli equation: Summary of results.

| Method | Training time (s) | RMSE | Relative $L^2$ error | architecture |
|---|---|---|---|---|
| PINN | $2303.71 \pm 278.68$ | $2.11\text{e-}3 \pm 4.79\text{e-}4$ | $4.21\text{e-}3 \pm 9.56\text{e-}4$ | $(2, 4\times 20, 1)$ |
| SWIM-ODE (our) | **0.05** | $6.06\text{e-}8 \pm 2.96\text{e-}8$ | $1.20\text{e-}7 \pm 5.91\text{e-}8$ | $(1, 200, 10, 1)$ |
| **ELM-ODE (our)** | 0.06 | **1.75e-8 $\pm$ 3.91e-9** | **3.50e-8 $\pm$ 7.79e-9** | $(1, 200, 10, 1)$ |
| *IGA-FEM* | 0.94 | 2.11e-7 | 4.21e-7 | 33 |

The training is performed on 1500 data points in the interior and boundary. We test the neural-PDE solvers with 5000 data points in the interior and boundary. The weights of the hidden layer for the ELM-ODE are sampled from the Gaussian distribution and biases from a uniform distribution in $[-1, 1]$. For our approach to handling time-dependent Dirichlet boundary conditions, please refer to Appendix B.1.1.

**Ablation studies:** For PINN, the results for the ablation studies for the width of the network and the number of data points are included in Table 12, Table 13. The ablation study for the number of neurons in the hidden layer of the network for ELM-ODE and SWIM-ODE is presented in Table 11. Additionally, we perform an ablation study for the SVD layer to demonstrate its impact on the computation time saved in Table 14. Particularly, we observe that with the SVD layer, the number of basis functions (width after the SVD layer) is reduced by up to 22x for ELM-ODE and up to 1.5x for SWIM-ODE and we obtain substantial speed-ups (more than a factor of 50) in the computation time.

**Comparison of results:** The exact architectures and comparison of training times and errors are presented in Table 10 and Table 15. Figure 13 shows the errors with all approaches and the ground truth.

## C.4 BURGERS

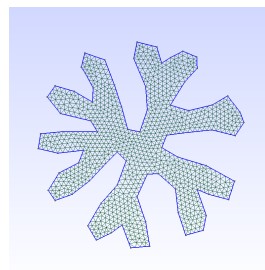
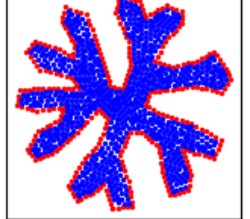

(a) Generated Mesh: FEM          (b) Sampled collocation points: Neural PDE solvers

Figure 12: Advatages of mesh-free methods: For mesh-based methods, a complicated mesh must be constructed (a), while for our method, one can easily sample arbitrary points on the domain (b).

Table 10: Non-linear diffusion equation: Summary of hyper-parameters.

|  | Parameter | Value |
| --- | --- | --- |
| SWIM-ODE | Number of hidden layers | 2 (nonlinear and SVD layer) |
|  | Hidden layer width | 500 |
|  | Activation | tanh |
|  | $L^2$-regularization | $10^{-15}$ |
|  | SVD cutoff | $10^{-15}$ |
|  | ODE solver tolerance | $10^{-6}$ |
|  | Loss | mean-squared error |
| ELM-ODE | Number of hidden layers | 2 (nonlinear and SVD layer) |
|  | Hidden layer width | 200 |
|  | Activation | tanh |
|  | $L^2$-regularization | $10^{-15}$ |
|  | SVD cutoff | $10^{-15}$ |
|  | ODE solver tolerance | $10^{-6}$ |
|  | Loss | mean-squared error |
| FEM | Number of entities | 154 |
|  | Number of nodes | 1193 |
|  | Number of elements | 2070 |
|  | Type of elements | Lagrange |
|  | Shape of elements | triangle |
|  | Degree of polynomials | 1 |
|  | Number of basis functions | 1193 |
|  | Solver | Newton solver |
|  | Timestep size | 0.001 |
| PINN | Number of hidden layers | 4 |
|  | Layer width | $[10, 20, \mathbf{30}, 40]$ |
|  | Activation | tanh |
|  | Optimizer | LBFGS |
|  | Epochs | 10000 |
|  | Loss | mean-squared error |
|  | Learning rate | 0.01 |
|  | Batch size | 1000 |
|  | Parameter initialization | Xavier (Glorot & Bengio, 2010) |
|  | Loss weights, $\lambda_1, \lambda_2$ | 0.01, 1 |
|  | # Interior points | $[8790, 1760, \mathbf{880}, 440]$ |
|  | # Initial and boundary points | $[3140, 630, \mathbf{320}, 160]$ |

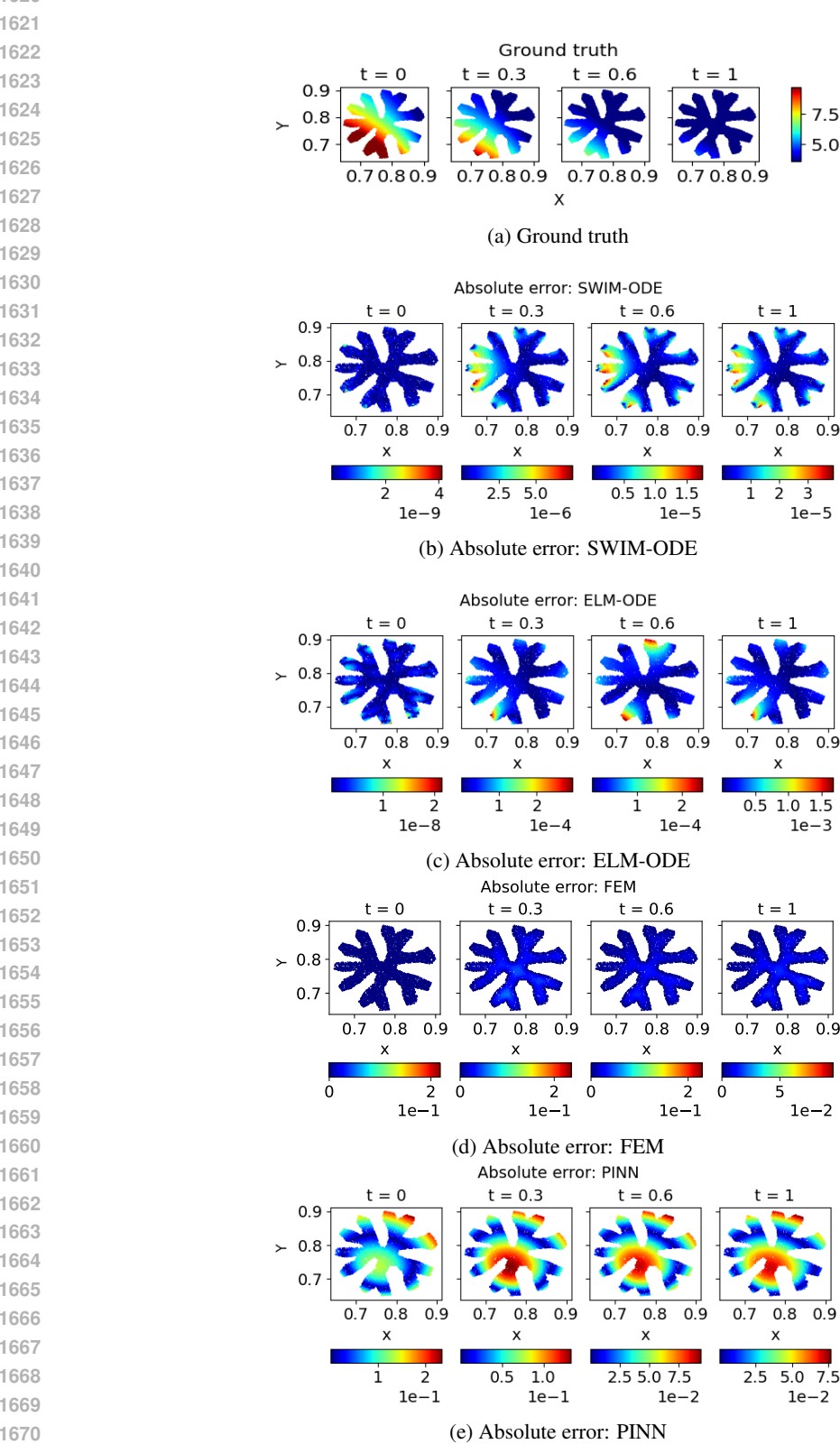

(a) Ground truth

(b) Absolute error: SWIM-ODE

(c) Absolute error: ELM-ODE

(d) Absolute error: FEM

(e) Absolute error: PINN

Figure 13: Non-linear diffusion equation: absolute error plots and ground truth at four-time instants

Table 11: Non-linear diffusion equation: ablation study for the network width for SWIM-ODE and ELM-ODE. The mean is computed over 3 seeds.

| Width | Relative $L^2$ error (SWIM-ODE) | Relative $L^2$ error (ELM-ODE) |
|---|---|---|
| 200 | 1.34e-4 | 4.92e-3 |
| 200 | 1.34e-4 | 4.92e-3 |
| 300 | 5.07e-6 | 3.13e-5 |
| 400 | 2.88e-6 | **1.02e05** |
| 500 | **3.02e-7** | 1.52e-5 |

Table 12: Non-linear diffusion equation equation 24: hyperparameter optimization for PINN varying layer width. The mean is computed over 3 seeds.

| Layer width | Training time (s) | RMSE | Relative $L^2$ error |
|---|---|---|---|
| 10 | **61.09 ± 1.62** | 4.11e-2 ± 2.04e-3 | 1.50e-2 ± 7.48e-4 |
| 20 | 68.05 ± 1.56 | 3.74e-2 ± 1.04e-3 | 1.37e-2 ± 3.82e-4 |
| 30 | 76.01 ± 0.57 | **3.67e-2 ± 1.03e-3** | **1.34e-2 ± 3.78e-4** |
| 40 | 82.43 ± 0.45 | 3.76e-2 ± 1.69e-3 | 1.37e-2 ± 6.21e-4 |

Table 13: Non-linear diffusion equation equation 24: hyperparameter optimization for PINN varying interior points

| Interior points | Training time (s) | RMSE | Relative $L^2$ error |
|---|---|---|---|
| 600 | **65.08 ± 4.23** | 3.74e-2 ± 1.04e-3 | 1.37e-2 ± 3.82e-4 |
| 1200 | 98.48 ± 3.78 | 3.51e-2 ± 6.67e-4 | 1.28e-2 ± 2.44e-4 |
| 2390 | 143.31 ± 5.50 | **3.34e-2 ± 6.53e-4** | **1.22e-2 ± 2.38e-4** |
| 11930 | 1154.48 | 2.01 | 0.73 |

Table 14: Non-linear diffusion equation: Ablation Study for the SVD layer with SWIM-ODE and ELM-ODE. We write $\infty$ if the computation takes more than 3 hours. ELM-ODE-accurate is the one that takes longer, but results in a much lower error, and ELM-ODE-fast is the one that takes less time but produces an error comparable/to or better than PINNs (which facilitates comparison with PINNs). We denote the ratio of the width of the hidden layer to the width of the SVD layer by $C_r$.

| Method | Quantity | With SVD layer | Without SVD layer | Ratio |
|---|---|---|---|---|
| ELM-ODE-accurate | Width | 62 | 300 | $C_r \approx 22.8$x |
| | Time (s) | 60.98 | 7087.38 | Speed-up $\approx 52$x |
| | Rel. $L_2$ error | 6.49e-8 | 1.02e-6 | - |
| ELM-ODE-fast | Width | 35 | 300 | $C_r \approx 8.5$x |
| | Time (s) | 30.57 | $\infty$ | Speed-up $\infty$ |
| | Rel. $L_2$ error | 5.12e-5 | - | - |
| SWIM-ODE | Width | 316 | 500 | $C_r \approx 1.5$x |
| | Time (s) | 328.03 | $\infty$ | Speed-up $\infty$ |
| | Rel. $L_2$ error | 2e-6 | - | - |

**Problem Setup:** The inviscid Burgers' equation is a non-linear PDE, which can form shock waves. We solve Burgers' equation on $\Omega = [-1, 1]$ for time $t \in (0, 1]$, so that

$$u_t + uu_x - (0.01/\pi)u_{xx} = 0, \quad x \in \Omega, \quad t \in [0, 1], \tag{27}$$

$$u(0, x) = -\sin(\pi x), \tag{28}$$

$$u(1, -1) = u(t, 1) = 0. \tag{29}$$

Table 15: Non-linear diffusion equation: Summary of results.

| Method | Training time (s) | RMSE | Relative $L^2$ error | architecture |
|---|---|---|---|---|
| PINN | 143.31 | 3.34e-2 ± 6.53e-4 | 1.22e-2 ± 2.38e-4 | (2, 4 × 30, 1) |
| ELM-ODE (our) | 30.57 | 5.02e-5 ± 4.84e-5 | 5.12e-5 ± 4.95e-5 | (2, 200, 1) |
| **SWIM-ODE (our)** | 423 | 1.96e-6 ± 1.95e-6 | 2.00e-6 ± 1.99e-6 | (2, 500, 1) |
| *FEM* | *2.71* | 7.33e-3 | 2.68e-3 | 1193 |

**Ablation studies:** The results of the ablation study with the number of neurons in the hidden layer for SWIM-ODE are presented in Table 16. We observe that starting with a width of 1200, the error decreases with a width up to 600 and increases again below 600. We believe that for widths lower than 600, the network capacity seems to be the reason for the loss of accuracy. For very high widths, the regularization constant has to be kept to a higher value to avoid overfitting. Otherwise, the ODE system becomes highly stiff and unstable. With this high regularization constant, the training becomes stable but affects the training accuracy. We do not include results for ELM-ODE as it fails on all widths as it is not able to capture the sharp shocks and exhibits Gibbs phenomenon Gottlieb & Shu (1997), which is explained in detail in Appendix C.4.1.

We describe additional details in solving the Burgers' equation with various neural PDE solvers in Table 17 and Table 18.

We also perform an ablation study for the SVD layer for SWIM-ODE. Please refer to Table 20. The ablation study reveals that the SVD layer compresses the number of neurons by a factor of 1.58, which reduces the output computation time by a factor of 7 for almost the same accuracy. This highlights the utility of the SVD layer.

**Comparison of results:** Figure 14 shows the absolute errors obtained with the PINN, Causal PINN, and IGA methods.

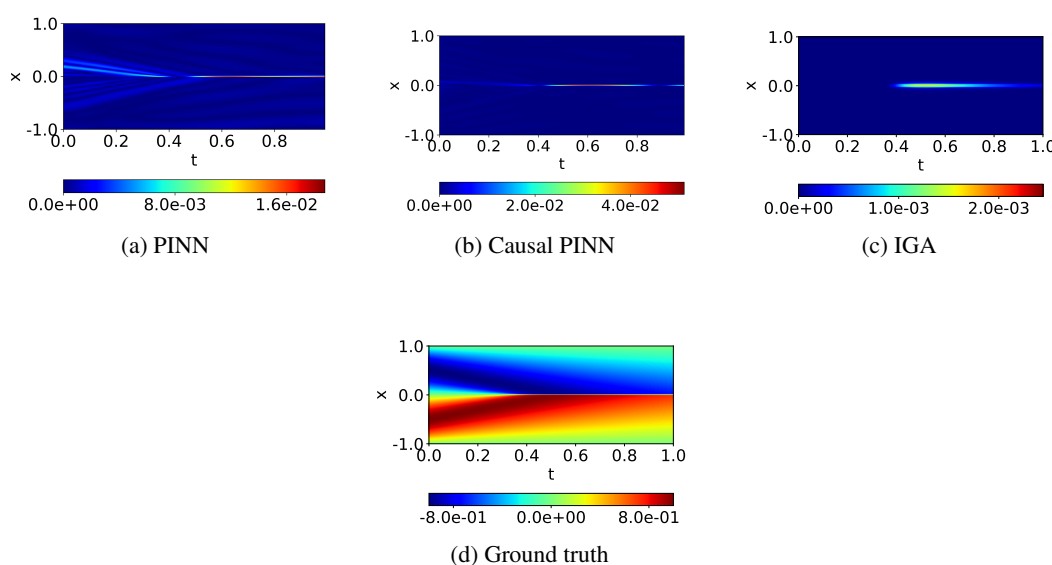

Figure 14: Burgers' equation: absolute error plots and ground truth

### C.4.1 COMPARISON WITH CLASSICAL SPECTRAL METHODS

We compare to other traditional spectral methods by fitting the Burgers' equation solution directly at a certain time step, where the function has a locally steep gradient. We argue that if a method fails to approximate this function well, it is unlikely to achieve better results in solving the PDE.

Table 16: Burgers' equation: ablation study for the network width for SWIM-ODE.

| Width | Relative $L^2$ error |
|---|---|
| 120 | 1.24e-2 |
| 240 | 4.27e-4 |
| **600** | **2.93e-4** |
| 800 | 3.12e-4 |
| 1200 | 4.92e-3 |

Table 17: Burgers' equation: Summary of hyper-parameters.

| | Parameter | Value |
|---|---|---|
| SWIM-ODE | Number of hidden layers | 2 |
| | Hidden layer width | $[450]$ |
| | Activation | tanh |
| | $L^2$-regularization | $[10^{-6}, \mathbf{10^{-7}}, 10^{-8}, 10^{-10}, 10^{-12}]$ |
| | Loss | mean-squared error |
| | # collocation points (space) | $[800]$ |
| | # sampling points | $[6000]$ |
| ELM-ODE | Number of hidden layers | 2 |
| | Hidden layer width | $[2000]$ |
| | Activation | tanh |
| | $L^2$-regularization | $[10^{-6}, \mathbf{10^{-7}}, 10^{-8}, 10^{-10}, 10^{-12}]$ |
| | Loss | mean-squared error |
| | # collocation points (space) | $[3000]$ |
| | # sampling points | $[6000]$ |

Figure 16 shows the approximation of the Burgers' equation solution at $t = 0.99$, using SWIM basis functions, ELM basis functions, Fourier series, and Chebyshev polynomials, respectively. The number of basis functions is 102 for all methods. Figure 15 shows the approximation error using a different number of basis functions. We can see that for ELM basis functions, Fourier series and Chebyshev polynomials, there are oscillations near the nonlinearity, and the error is large compared to the SWIM basis functions, where we are able to take the advantage of resampling data points and basis functions in order to adapt to the target function well. Note that in this experiment, the weights for the ELM basis functions are sampled from a Gaussian distribution with a standard deviation of 10 in order to increase the number of basis functions. The biases are sampled from a uniform distribution in $[-10, 10]$. For the Fourier series and Chebyshev polynomials, we use equispaced grid points. We also experimented with quadrature points and placed more points near the steep gradient in an attempt to mitigate the oscillations associated with the Gibbs phenomenon and the Runge phenomenon, but it did not lead to any significant improvement in the results.

## C.5 HIGH-DIMENSIONAL DIFFUSION EQUATION

**Problem setup:** We consider up to 10-dimensional diffusion equation on spatial domain $\Omega = [-1, 1]^d$ and time domain $t \in (0, 1]$:

$$u_t - \Delta u = \left(\frac{1}{d} - 1\right) \cos\left(\frac{1}{d} \sum_{i=1}^{d} x_i\right) \exp(-t), \quad x \in \Omega, \quad t \in [0, 1], \tag{30}$$

$$u(x, t) = \cos\left(\frac{1}{d} \sum_{i=1}^{d} x_i\right) \exp(-t), \tag{31}$$

$$u(x, 0) = \cos\left(\frac{1}{d} \sum_{i=1}^{d} x_i\right), \tag{32}$$

Table 18: Burgers' equation: Network hyper-parameters used for PINN, Causal PINN, and IGA.

|  | Parameter | Value |
|---|---|---|
| PINN | Number of hidden layers | 9 |
|  | Layer width | 20 |
|  | Activation | tanh |
|  | Optimizer | LBFGS |
|  | Epochs | 10000 |
|  | Loss | mean-squared error |
|  | Learning rate | 0.1 |
|  | Batch size | 200 |
|  | Parameter initialization | Xavier (Glorot & Bengio, 2010) |
|  | Loss weights, $\lambda_1, \lambda_2$ | 1, 1 |
|  | # Interior points | 10000 |
|  | # Initial and boundary points | 600 |
| Causal PINN | Number of hidden layers | 9 |
|  | Layer width | 20 |
|  | Activation | tanh |
|  | Optimizer | ADAM followed by LBFGS |
|  | ADAM Epochs | 5000 |
|  | LBFGS Epochs | 10000 |
|  | Loss | mean-squared error |
|  | Learning rate | 0.1 |
|  | Batch size | 200 |
|  | Parameter initialization | Xavier (Glorot & Bengio, 2010) |
|  | Loss weights, $\lambda_1, \lambda_2$ | 1, 1 |
|  | # Interior points | 40000 |
|  | # Initial and boundary points | 600 |
|  | Causality parameter, $\epsilon$ | 5 |
| IGA | Number of nodes | 400 |
|  | Degree of polynomials | 8 |
|  | Number of basis functions | 405 |

Table 19: Burgers' Equation: Summary of results.

| Method | Training time (s) | RMSE | Relative $L^2$ error | architecture |
|---|---|---|---|---|
| ELM-ODE (our) | 2.41 | 1.51e-1 $\pm$ 3.27 e-4 | 2.47e-1 $\pm$ 5.33e-4 | (1, 2000, 1) |
| PINN | 275.23 $\pm$ 5.38 | 2.38e-3 $\pm$ 1.61e-3 | 3.88e-3 $\pm$ 2.61e-3 | (2, 9$\times$20, 1) |
| Causal-PINN | 1531.79 $\pm$ 18.45 | 9.85e-3 $\pm$ 5.51e-3 | 1.60e-2 $\pm$ 8.97e-3 | (2, 9$\times$20, 1) |
| **SWIM-ODE (our)** | **141.35** | **2.05e-4 $\pm$ 2.84e-4** | **3.33e-4 $\pm$ 4.63e-4** | (1, 400, 1) |
| *IGA-FEM* | 13.61 | 1.35e-4 | 2.20e-4 | 405 |

Table 20: Burgers' Equation: Ablation Study for the SVD layer with SWIM-ODE.

|  | With SVD layer | Without SVD layer | Ratio |
|---|---|---|---|
| Number of neurons | 500 | 316 | Width Compression $\approx 1.58$x |
| Time (s) | 141.5 | 989.84 | Speed-up $\approx 7$x |
| Rel. $L_2$ error | 3.34e-4 | 3.28e-4 | - |

where $d \in \{3, 5, 7, 10\}$. This example is considered in (Wang & Dong, 2024), but only up to 7 dimensions. For the high-dimensional diffusion equation, we use 16000 training points in the interior and 4000 points on the boundary randomly sampled using the Latin hypercube strategy. The test data contains 8000 points in the interior and 2000 points on the domain's boundary, which were also sampled with a Latin hypercube strategy.

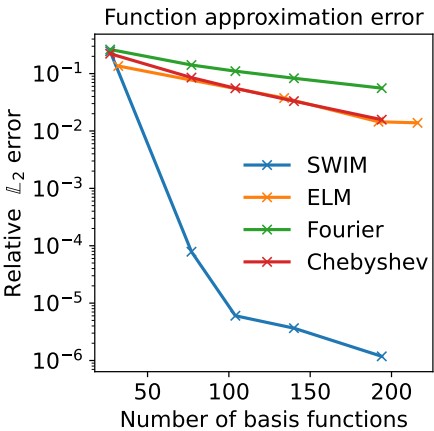

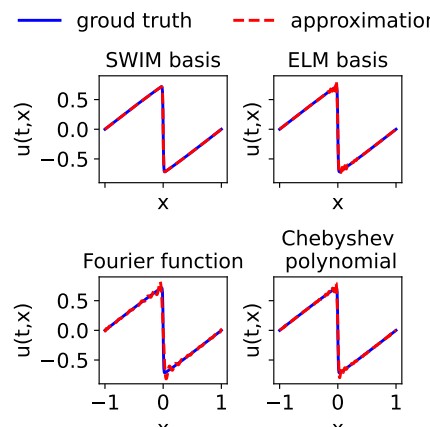

Figure 15: Approximation error for four types of basis functions. Here, we directly fit the Burgers' equation solution at $t = 0.99$. The approximation error decreases as we increase the number of basis functions, and the SWIM basis functions yield the best result among all methods.

Figure 16: Approximation of Burgers' equation solution at $t = 0.99$ with four types of basis functions. The number of basis functions in all cases is 102. Oscillations can be seen near the steep gradient for the methods using ELM basis functions, Fourier functions, and Chebyshev polynomials.

**Ablation studies:**    The ablation study with respect to the network width for ELM-ODE and SWIM-ODE is already presented in Figure 7, where we observe a rapid exponential decay of error with respect to increasing width of the network (even exponential convergence for the high-dimensional diffusion equation in 3 and 5 dimensions).

The hyper-parameters for all neural PDE solvers considered in this work are presented in Table 21, and the results for the high-dimensional diffusion equation with up to 3, 5, 7, and 10 dimensions are summarized in Table 22.

We also perform an ablation study on the SVD layer. To quantify the compression in width after the SVD layer, we define a compression ratio as $C_r = \frac{M_s}{r}$, where $M_s$ is the width of the (sampled) hidden layer before the SVD layer (see Figure 3), and $r$ is the width of the SVD layer. We define a speed-up in computation time as $s = \frac{T_{\text{no-svd}}}{T_{\text{svd}}}$ as the ratio of computational time without the SVD layer to the time required with the SVD layer.

The results of the ablation study for the SVD layer with the high-dimensional diffusion equations demonstrate that for ELM-ODE, the SVD layer results in substantial speed-ups for 3, 5, and 7 dimensional heat equations - by factors of 52, 77, and 21 respectively. We observe that the compression ratios achieved with the SVD layer are also substantial 22.8, 5, and 1.2, for dimensions 3, 5, and 7, respectively. For the 10-dimensional diffusion equation, to cover the high-dimensional space, we observe a (relatively lower compared to other dimensions) compression ratio of 1.4, as more basis functions are required to represent functions in high dimensions accurately. Thus, the time required with the SVD layer is around 94 percent of the time required without the SVD layer. In all the cases, the loss is always in the same order as the one without the SVD layer.

Note that in all cases, the extra cost of computing the SVD easily pays off by substantially saving time in the ODE solver for ELM-ODE. This is because of the improved conditioning of the feature matrix and the reduction in the size of the ODE system to be solved. With SWIM-ODE, the observations are similar but with lower compression ratios and speed-ups. But, for this problem, SWIM-ODE is not the preferred method, as the underlying solution is smooth, has low-frequency spatial variations, and does not have steep gradients anywhere in the domain. Thus, SWIM basis functions are not optimal in the vanilla setting. See Appendix B.1.3 for details on this.

**Comparison of results:**    We demonstrate that ELM-ODE accurately solves the high-dimensional diffusion equation by visualizing the ground truth, the ELM-ODE solution, and the point-wise

absolute error for the 10-dimensional diffusion equation across different cross-sections for a fixed time in Figure 17 and the time evolution of solution at some sampled points in space in Figure 18. We show the solution across different spatial coordinates evaluated at three different time instants (rest of the coordinates fixed to the center) in Figure 19.

Table 21: Summary of hyper-parameters for the 10-dimensional diffusion equation.

|  | Parameter | Value |
|---|---|---|
| SWIM-ODE | Number of hidden layers | 2 (nonlinear and SVD layer) |
|  | Hidden layer width | 4000 |
|  | Activation | tanh |
|  | $L^2$-regularization | $10^{-10}$ |
|  | SVD cutoff | $10^{-10}$ |
|  | ODE solver tolerance | $10^{-6}$ |
|  | Loss | mean-squared error |
| ELM-ODE-fast | Number of hidden layers | 2 (nonlinear and SVD layer) |
|  | Hidden layer width | 1000 |
|  | Activation | tanh |
|  | $L^2$-regularization | $10^{-10}$ |
|  | SVD cutoff | $10^{-10}$ |
|  | ODE solver tolerance | $10^{-6}$ |
|  | parameter range $[-r_m, r_m]$ | $r_m = 0.05$ |
|  | Loss | mean-squared error |
| ELM-ODE-accurate | Number of hidden layers | 2 (nonlinear and SVD layer) |
|  | Hidden layer width | 4000 |
|  | Activation | tanh |
|  | $L^2$-regularization | $10^{-10}$ |
|  | SVD cutoff | $10^{-10}$ |
|  | ODE solver tolerance | $10^{-6}$ |
|  | parameter range $[-r_m, r_m]$ | $r_m = 0.05$ |
|  | Loss | mean-squared error |
| PINN | Number of hidden layers | 4 |
|  | Layer width | 20 |
|  | Activation | tanh |
|  | Optimizer | LBFGS |
|  | Epochs | 1000 |
|  | Loss | mean-squared error |
|  | Learning rate | 0.1 |
|  | Batch size | 4000 |
|  | Parameter initialization | Xavier (Glorot & Bengio, 2010) |
|  | Loss weights, $\lambda_1, \lambda_2$ | 1, 1 |
|  | # Interior points | 16000 |
|  | # Initial and boundary points | 4000 |

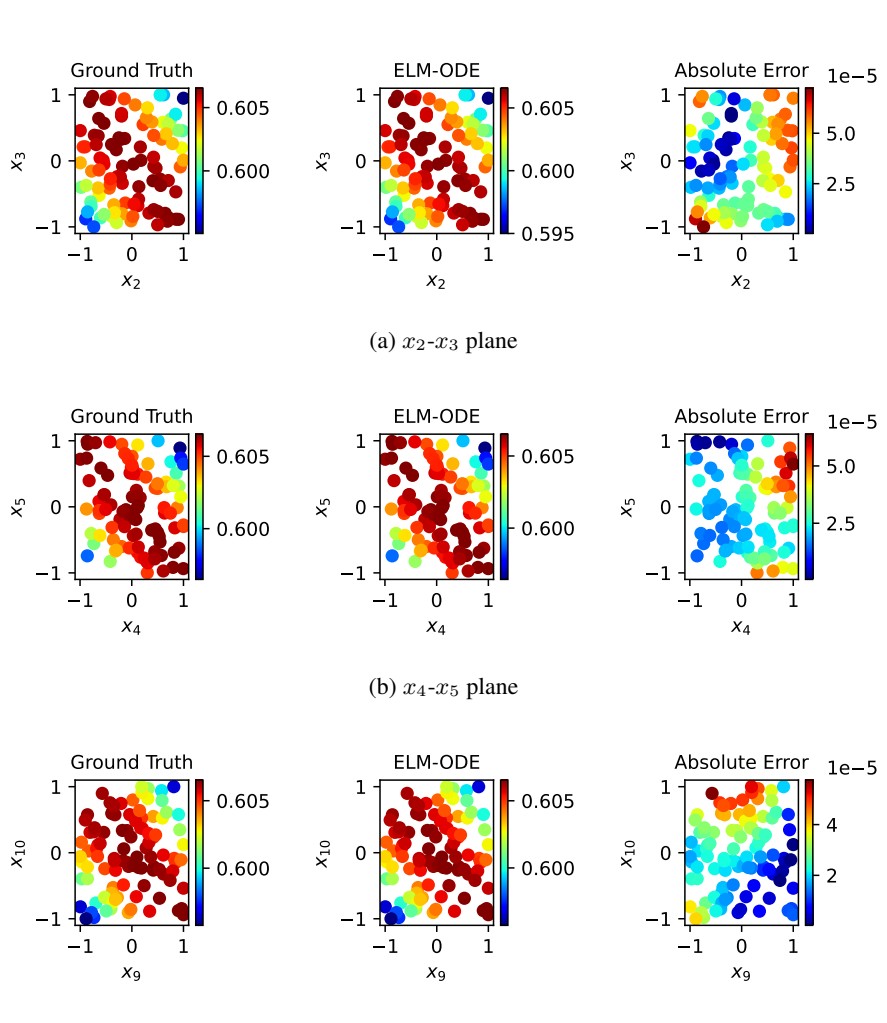

(a) $x_2$-$x_3$ plane

(b) $x_4$-$x_5$ plane

(c) $x_9$-$x_{10}$ plane

Figure 17: 10-dimensional diffusion equation: Ground truth, ELM-ODE solution, and point-wise absolute error across different cross-sections of the spatiotemporal domain (located exactly in the center of the spatial-temporal domain with respect to the remaining coordinates).

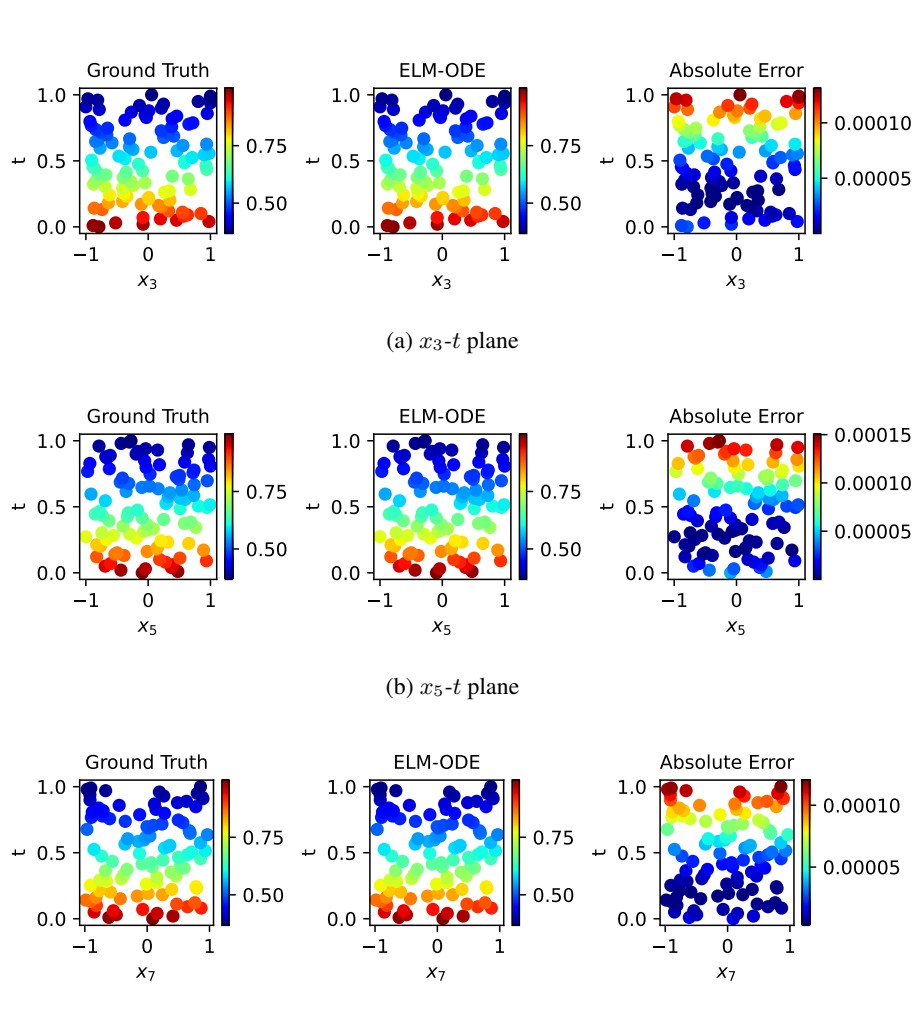

(a) $x_3$-$t$ plane

(b) $x_5$-$t$ plane

(c) $x_7$-$t$ plane

Figure 18: 10-dimensional diffusion equation: Ground truth, ELM-ODE solution, and point-wise absolute error at various planes at different time points (The rest of the spatial coordinates are set to the center of the spatial-temporal domain).

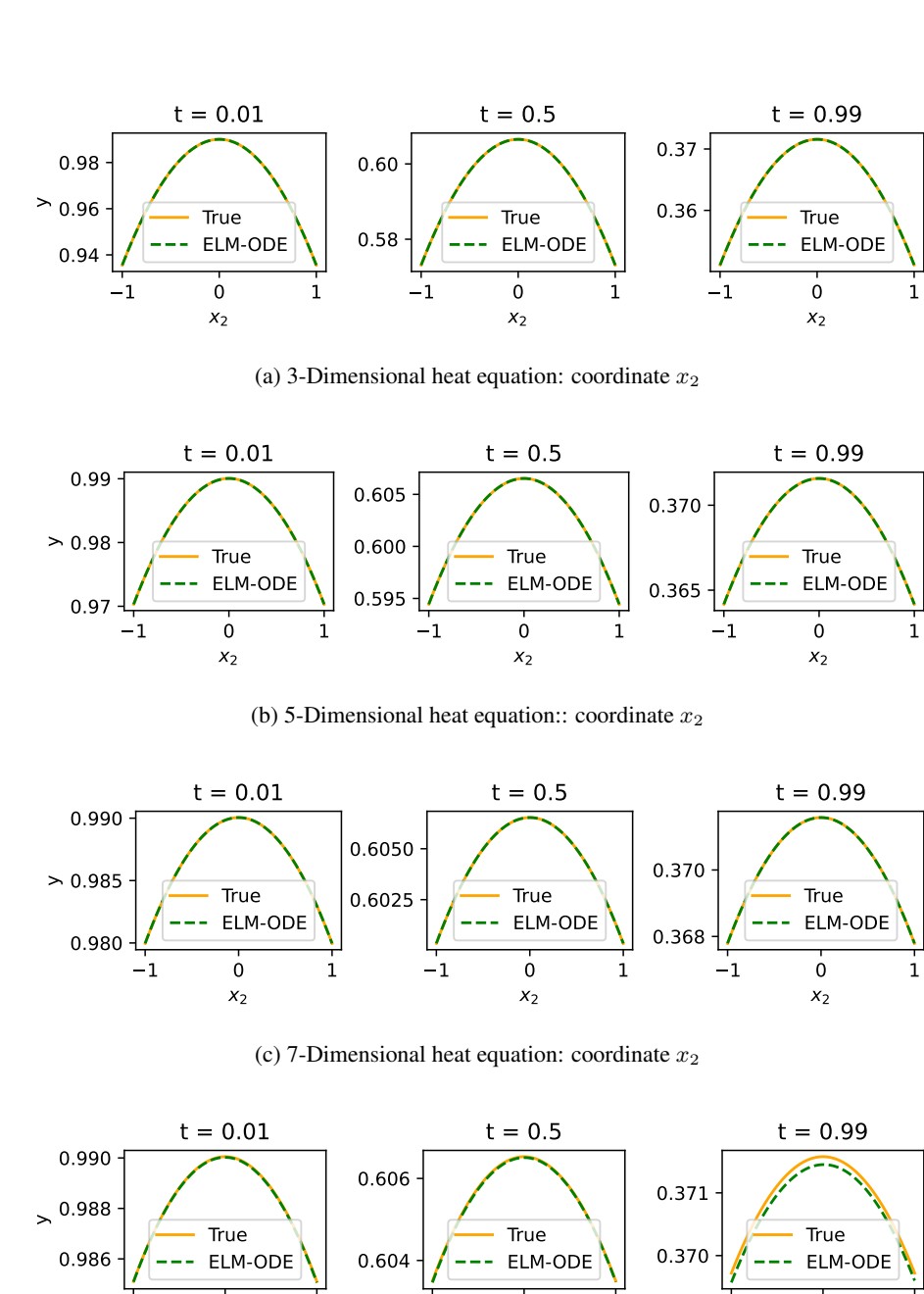

(a) 3-Dimensional heat equation: coordinate $x_2$

(b) 5-Dimensional heat equation:: coordinate $x_2$

(c) 7-Dimensional heat equation: coordinate $x_2$

(d) 10-Dimensional heat equation: coordinate $x_2$

Figure 19: High-dimensional diffusion equation: Ground truth and ELM-ODE solution across different spatial coordinates evaluated at $t = 0.01$ (column 1), $t = 0.5$ (column 2) and $t = 0.99$ (column 3), where the rest of the spatial coordinates are set to the center of the spatial-temporal domain.

Table 22: Summary of results for high-dimensional diffusion equation.

| Dimension | Method | Time (s) | RMSE | Relative $L^2$ error |
|---|---|---|---|---|
| 3-d | PINN | 102.32 | 2.84e-4 $\pm$ 3.73e-5 | 4.54e-4 $\pm$ 5.97e-5 |
| | SWIM-ODE (our) | 95.73 | 2.18e-6 $\pm$ 1.93e-6 | 5.37e-6 $\pm$ 4.27e-7 |
| | **ELM-ODE-fast (our)** | **0.9** | 2.42e-6 $\pm$ 1.37e-6 | 3.90e-6 $\pm$ 2.98e-6 |
| | **ELM-ODE-accurate (our)** | 60.98 | **3.48e-8 $\pm$ 2.17e-6** | **6.49e-8 $\pm$ 4.31e-8** |
| 5-d | PINN | 133.95 | 2.91e-4 $\pm$ 5.34e-5 | 4.52e-4 $\pm$ 8.30e-5 |
| | SWIM-ODE (our) | 129.65 | 1.03e-4 $\pm$ 5.94e-5 | 2.39e-4 $\pm$ 8.69e-5 |
| | **ELM-ODE-fast (our)** | **1.2** | 1.25e-4 $\pm$ 2.42e-5 | 3.74e-4 $\pm$ 5.37e-5 |
| | **ELM-ODE-accurate (our)** | 102.95 | **4.71e-7 $\pm$ 3.56e-7** | **7.5e-7 $\pm$ 3.92e-7** |
| 7-d | PINN | 163.89 | 3.05e-4 $\pm$ 2.94e-5 | 4.69e-4 $\pm$ 4.51e-5 |
| | SWIM-ODE (our) | 198.20 | 3.96e-4 $\pm$ 1.03e-4 | 7.8e-4 $\pm$ 2.50e-4 |
| | **ELM-ODE-fast (our)** | **5.95** | 1.05e-5 $\pm$ 8.76e-6 | 2.21e-5 $\pm$ 1.01e-5 |
| | **ELM-ODE-accurate (our)** | 176.95 | **1.19e-6 $\pm$ 2.93e-7** | **2.54e-6 $\pm$ 5.10e-7** |
| 10-d | PINN | 189.67 | 3.98e-4 $\pm$ 6.59e-5 | 6.06e-4 $\pm$ 1.00e-4 |
| | SWIM-ODE (our) | 61.07 | 1.01e-3 $\pm$ 3.09e-4 | 2.31e-3 $\pm$ 1.03e-3 |
| | **ELM-ODE-fast (our)** | **2.07** | 2.89e-4 $\pm$ 5.91e-5 | 4.46e-4 $\pm$ 9.61e-5 |
| | **ELM-ODE-accurate (our)** | 182.91 | **1.04e-5 $\pm$ 3.32e-6** | **2.28e-5 $\pm$ 5.91e-6** |

Table 23: High-dimensional diffusion equation: Ablation Study for the SVD layer with SWIM-ODE.

| Dimension | Quantity | With SVD layer | Without SVD layer | Ratio |
|---|---|---|---|---|
| 3-d | Width | 1391 | 4000 | Compression $\approx$ 2.9x |
| | Time (s) | 95.73 | 388.12 | Speed-up $\approx$ 4x |
| | Rel. $L_2$ error | 5.29e-6 | 4.77e-6 | - |
| 5-d | Width | 1437 | 4000 | Compression $\approx$ 2.8x |
| | Time (s) | 129.65 | 199.92 | Speed-up $\approx$ 1.5x |
| | Rel. $L_2$ error | 2.39e-4 | 2.18e-4 | - |
| 7-d | Width | 3114 | 4000 | Compression $\approx$ 1.3x |
| | Time (s) | 120.32 | 198.31 | Speed-up $\approx$ 1.6x |
| | Rel. $L_2$ error | 7.83e-4 | 7.83e-4 | - |
| 10-d | Width | 3100 | 4000 | Compression $\approx$ 1.3x |
| | Time (s) | 121.93 | 111.8 | Speed-up $\approx$ 0.91x |
| | Rel. $L_2$ error | 2.30e-3 | 2.30e-3 | - |

Table 24: High-dimensional diffusion equation: Ablation Study for the SVD layer with ELM-ODE.

| Dimension | Quantity | With SVD layer | Without SVD layer | Ratio |
|---|---|---|---|---|
| 3-d | Width | 175 | 4000 | Compression $\approx$ 22.8x |
| | Time (s) | 60.98 | 7087.38 | Speed-up $\approx$ 52x |
| | Rel. $L_2$ error | 6.49e-8 | 1.02e-6 | - |
| 5-d | Width | 794 | 4000 | Compression $\approx$ 5x |
| | Time (s) | 89.27 | 6873.8 | Speed-up $\approx$ 77x |
| | Rel. $L_2$ error | 7.30e-7 | 2.19e-6 | - |
| 7-d | Width | 3336 | 4000 | Compression $\approx$ 1.2x |
| | Time (s) | 176.95 | 3770.09 | Speed-up $\approx$ 21x |
| | Rel. $L_2$ error | 2.54e-6 | 4.06e-6 | - |
| 10-d | Width | 2856 | 4000 | Compression $\approx$ 1.4x |
| | Time (s) | 119 | 127 | Speed-up $\approx$ 1.06x |
| | Rel. $L_2$ error | 5.57e-5 | 4.36e-5 | - |

