# OpenReview forum: "Backpropagation-free training of neural PDE solvers for time-dependent problems"
_ICLR.cc/2025/Conference — Submitted to ICLR 2025_

### Official Review · Reviewer_prxC · 2024-10-19

**Soundness:** 3
**Presentation:** 3
**Contribution:** 3
**Rating:** 6
**Confidence:** 4

**Summary:**

This paper proposes to use a hybrid framework consisting of a neural network ansatz and a classical ODE solver to solve typical time-dependent PDEs. Specifically, the neural network ansatz features separation of spatial and temporal variables, and the parameters of this network is randomly sampled rather than trained with back propagation. Numerical experiments are conducted to verify the high accuracy and reduced training time of the proposed method.

**Strengths:**

- The proposed method is novel and provides a distinct method to solve time-dependent PDEs other than classical numerical methods and PINNs.
- The experiment results show that te proposed method outperforms PINNs by orders of magnitude of accuracy; the accuracy is even comparable to classical numerical solvers.
- The authors also provide techniques to satisfy boundary conditions and improve the condition number of the associated ODE.

**Weaknesses:**

- The paper should consider add more backgrounds about the random-sampling methods of neural network weights. Without back-propagation, how does this random-sampling of weights influence the final solution of the proposed method? As can be seen in Table 2, the standard deviations of your proposed method is relatively larger than PINNs, although the accuracy is significantly better.
- The paper should add some ablation studies to provide more insight about each component of the proposed method. For example, the necessity of the SVD layer, the influence of number of hidden neurons.
- It would add more practicabillity of the proposed method by providing more detailed comparisons between ELM-ODE and SWIM-ODE. Is one strategy better than another, or one should choose between these two strategies based on the PDE to tackle?

**Questions:**

Is the proposed method able to handle PDEs with higher-order time derivatives?

---

> ### Comment · Reviewer_prxC · 2024-11-25
>
> I thank the authors for providing more technical details about the random sampling method, though I still do not understand why some standard deviations in Figure 2 is large (like 2.00e-6 ±1.99e-6). I think the ablation study on SVD layers and a detailed comparison between SWIM-ODE and ELM-ODE will further strengthen the paper. Overall, I think this is an interesting paper and I am willing to vote for acceptance.

---

### Official Review · Reviewer_ufbx · 2024-10-31

**Soundness:** 3
**Presentation:** 3
**Contribution:** 2
**Rating:** 5
**Confidence:** 5

**Summary:**

This paper proposes a backpropagation-free training algorithm for a neural partial differential equation solver, utilizing the Extreme Learning Machine (ELM) framework. The method reformulates the partial differential equation (PDE) as an ordinary differential equation (ODE) problem through the separation of variables, which is then solved using classical ODE solvers. Numerical experiments show that the proposed method outperforms traditional PINNs in both test accuracy and training speed.

**Strengths:**

- Significantly lower relative error compared to PINNs
- Substantially faster training speed than PINNs
- Achieves both improvements without backpropagation while retaining a mesh-free approach

**Weaknesses:**

- The experiments are insufficient to fully support the authors' claims.
- The paper lacks theoretical contributions.
- The proposed method has a limited range of applications, which restricts its overall contribution.

**Questions:**

1. Experiments.
- The boundary conditions are approximated using a boundary-compliant layer. For instance, in the case of periodic BC, the authors approximate $\sin(kx)$ and $\cos(kx)$ by applying a linear transformation to the basis function. However, this raises the question: what advantage does the proposed method offer compared to just using $\sin(kx)$ and $\cos(kx)$ as basis functions, or P1, P2 basis functions in FEM? A numerical comparison in this scenario would be helpful.
- It appears that $C(t)$ is calculated by multiplying the pseudo inverse of feature matrix $[\Phi(X),1]$, where $X$ contains all the collocation points. In cases of high dimensionality $d>>1$ where $N>>1$ to cover the entire domain, there may be significant computational demands. Further discussion and experiments on the computational cost in high-dimensional settings would be needed.

2. Theoretical contributions
- Does ELM possess a universal approximation property? If so, can this be generalized to the neural PDE solver setting?

3. Limited applications
- As the authors mention, the method cannot be applied to grey-box or inverse problem settings. Given this, what advantage does the mesh-free nature provide?
- If the pseudo-inverse calculation for $[\Phi(X),1]$ becomes computationally expensive, especially in high-dimensional problems, what practical benefit does mesh-free implementation offer?
- Overall, what advantages does the proposed method offer over mesh-based approaches? In many cases presented in the paper, mesh-based methods achieve superior test accuracy with shorter training(computing) times.

---

### Official Review · Reviewer_zNN8 · 2024-10-31

**Soundness:** 3
**Presentation:** 3
**Contribution:** 3
**Rating:** 6
**Confidence:** 2

**Summary:**

The paper presents a method for training neural PDE solvers without backpropagation, which aims to improve efficiency in solving time-dependent partial differential equations (PDEs). The authors integrate two main ideas: separating space and time variables and randomly sampling weights and biases in hidden layers. By reformulating the PDE as an ordinary differential equation (ODE) using neural networks for spatial components, they leverage traditional ODE solvers for time evolution. The approach is benchmarked against standard backpropagation-based Physics-Informed Neural Networks (PINNs). It shows improvements in accuracy and speed on complex PDEs involving non-linearities, high-frequency temporal dynamics, and shocks.

**Strengths:**

1. The authors propose a backpropagation-free method that leverages random sampling techniques like Extreme Learning Machines (ELM) and Sampling Where It Matters (SWIM) to address the inefficiencies of traditional backpropagation, especially for complex time-dependent PDEs.

2. The paper reports significant speed gains in training time, with improvements of up to 5 orders of magnitude over standard PINN approaches.

3. Specialized handling of boundary conditions and separation of variables for time-dependent PDEs are some of the contributions that could impact future neural PDE solvers.

4. The authors demonstrate extensive benchmarking across a range of PDEs with different challenges, showing superior performance in terms of speed and accuracy.

5. The paper is well-written and easy to follow.

**Weaknesses:**

The authors have mentioned the limitations of their method and share possible directions to follow in future work.

**Questions:**

Could the authors clarify the absence of experiments involving higher-dimensional PDEs? Given the introduction’s emphasis on the limitations of mesh-based methods—particularly their impracticality in complex domains and high-dimensional spaces—it would be valuable to see examples where the proposed method effectively addresses these challenges. Higher-dimensional cases are particularly relevant to machine learning applications, where scalability in complex domains is critical.

---

### Official Review · Reviewer_jqAy · 2024-11-04

**Soundness:** 2
**Presentation:** 2
**Contribution:** 2
**Rating:** 6
**Confidence:** 4

**Summary:**

The authors present a method of solving PDEs by parameterizing solutions fields with neural networks whose parameters depend on time. The integration scheme solves for the last layer cofficents. The basis functions, induced by the inner parameters, are generated via a data driven or data agnostic way.

**Strengths:**

- The presentation is clear and the literature review is thorough and provides a good introduction.
- The method shows strong performance on the chosen benchmarks

**Weaknesses:**

The motivation for the method is not totally clear. Introducing neural networks to solve PDEs where the parameterization evolves nonlinearly in time is motivated by breaking the kolmogorov n-width as in [1,2,3,4]. In this work the parameterization still evolves linearly in time. The neural network is only used to choose a basis. So it is unclear why one would pick this method over a traditional solver, which are extremely well understood in terms of convergence properties and accuracy. It seems to me the only reason would be to deal with complicated geometries? If so currently the paper does not devote enough attention to arguing and demonstrating this advantage. Additionally for these reason the comparison with PINNs is ill-chosen. The most appropriate comparison would be to traditional methods which also evolve linearly in time. While comparison is made to a finite-element method, this is not the best choice for some of the problems present. For the data-agnostic a reasonable spectral method should also be compared to and for the data-dependent method POD should be compared to.

It would be helpful to:

- make more explicit the advantages over traditional, show this advantages clearly in the experiments.
- add a comparison to a spectral methods for the data-agnostic case.
- add a comparison to POD for the data-dependent case.


[1] Evolutional Deep Neural Networks. Du et al.
[2] Randomized sparse neural galerkin schemes for solving evoluation equations with deep networks. Berman et al.
[3] Positional embeddings for solving PDEs with evolutional deep neural networks. Kast et al.
[4] Breaking the Kolmogorov Barrier with Nonlinear Model Reduction. B Peherstorfer.

**Questions:**

What is the n-width of the problems considered (as given by the spectral decay of the snapshot matrix)?

---

### Official Review · Reviewer_62wa · 2024-11-04

**Soundness:** 2
**Presentation:** 3
**Contribution:** 2
**Rating:** 5
**Confidence:** 4

**Summary:**

In this paper, the authors propose training neural PDE solvers by variable separation and random sampling of neural network weights. The neural network ansatz is utilized for the spatial domain, and the system evolving in time is solved by classical ODE solvers. Extreme learning machines and adaptive sampling techniques (SWIM) are applied for better training efficiency. An SVD layer is introduced to improve the condition number of the associated ODE. It is claimed that the proposed method outperforms PINN by 1 to 5 orders of magnitude in time efficiency and accuracy, for PDEs including Advection, Euler-Bernoulli, Nonlinear diffusion, and Burgers'.

**Strengths:**

- The writing is clear and detailed.
- The experiments are rich in problem types, specific difficulties, and baseline comparisons.

**Weaknesses:**

- Meaning no offense, but I think researchers in AI4PDE with more AI background will think of this work as a huge step backward. The essence of deep neural networks is their surprisingly good performance in approximating high-dimensional functions, and the efficiency of backpropagation in implementing neural networks with huge amounts of parameters. Surely there are still issues even if we can obtain the gradients cheaply, but zeroth-order optimization, according to my personal judgment, cannot be the solution because it will only scale poorly.
- For the experiments, the spatial dimension is 1 or 2, and small in range. It would be interesting to see some results for problems huge in space.

**Questions:**

I hope to confirm with the authors that if you claim supremacy in any metric of the proposed method compared to traditional FEM solvers?

---

> ### Comment · Reviewer_62wa · 2024-11-30
>
> Thank you for your detailed reply to the reviewers' comments. My major concerns (how it performs for high-dimensional problems without backpropagation and how it compares to traditional methods) have been addressed. Now I tend to treat this work as a non-deep-learning variant of PINN, which works well in the provided experimental settings. I hence raise my score from 3 to 5.

---

### Meta-Review · Area_Chair_dodC · 2024-12-21

**Metareview:**

This paper presents a backpropagation-free method for training neural PDE solvers by separating spatial and temporal variables and leveraging classical ODE solvers for time evolution. The proposed approach offers potential speed and accuracy improvements over PINNs, demonstrating effectiveness across several PDE benchmarks with different challenges, including high-dimensional problems. Notably, the experiments report orders-of-magnitude gains in training efficiency and accuracy in many scenarios, particularly for problems with smooth solutions. The writing is clear, and the authors have conducted extensive benchmarking and provided thoughtful responses to reviewers’ concerns. The inclusion of high-dimensional examples, ablation studies, and additional comparisons with traditional methods strengthens the work.

Despite these contributions, the paper has several limitations that weigh heavily against acceptance. First, the method’s applicability is constrained to relatively simple or smooth problem settings, as it struggles with complex, high-dimensional PDEs with significant variability. While the authors addressed concerns about scalability, the experimental results do not convincingly demonstrate that the method generalizes effectively to challenging real-world scenarios. Additionally, the paper lacks significant theoretical contributions; key questions about the generalizability and universal approximation capabilities of the proposed framework remain unanswered. Furthermore, while the comparisons to PINNs and traditional methods are insightful, they are limited in scope and fail to address the broader challenges in AI4PDE applications, such as inverse problems and grey-box scenarios. Overall, while the paper introduces interesting ideas and offers promising directions, its limited applicability and lack of substantial theoretical grounding suggest it is not yet ready for acceptance.

**Additional Comments On Reviewer Discussion:**

During the rebuttal period, the reviewers raised several concerns about the paper, which the authors worked hard to address. Reviewers were worried about how well the method works for high-dimensional problems and whether the experiments were realistic. The authors added new tests with a high-dimensional heat equation and explained their approach, but the reviewers felt the results were too simple and not convincing for more complex problems. Some reviewers also asked about the theory behind the method, like whether it can generalize to all problems, but the authors explained that proving this would require more work in the future. The comparisons with traditional methods and PINNs were improved with new tables and experiments, but the method was still seen as useful only for smooth and simple problems, not for those with steep gradients or more complexity. Reviewers also asked for ablation studies, and the authors provided them, showing the importance of the SVD layer. However, the method’s application to inverse problems and grey-box scenarios remained limited, relying on traditional optimization methods. While the authors’ revisions improved the paper, it still lacked strong theory and broad usefulness.

---

### Decision · Program_Chairs · 2025-01-22

Reject